# Multi-Agent Teams Hold Experts Back

**Aneesh Pappu**[1]   **Batu El**[1]   **Hancheng Cao**[2]   **Carmelo di Nolfo**[3]   **Yanchao Sun**[3]   **Meng Cao**[3]   **James Zou**[1]

## Abstract

Multi-agent LLM systems are increasingly deployed as autonomous collaborators, where agents interact freely rather than execute fixed, pre-specified workflows. In such settings, effective coordination cannot be fully designed in advance and must instead emerge through interaction. However, most prior work enforces coordination through fixed roles, workflows, or aggregation rules, leaving open the question of how well self-organizing teams perform when coordination is unconstrained. Drawing on organizational psychology, we study whether self-organizing LLM teams achieve *strong synergy*, where team performance matches or exceeds the best individual member. Across human-inspired and frontier ML benchmarks, we find that—unlike human teams—LLM teams consistently fail to match their expert agent's performance, even when explicitly told who the expert is, incurring performance losses of up to 41.1% on ML benchmarks. Decomposing this failure, we show that expert leveraging, rather than identification, is the primary bottleneck. Conversational analysis reveals a tendency toward integrative compromise—averaging expert and non-expert views rather than appropriately weighting expertise—which increases with team size and correlates negatively with performance. Interestingly, this consensus-seeking behavior improves robustness to adversarial agents, suggesting a trade-off between alignment and effective expertise utilization. Our findings reveal a significant gap in the ability of self-organizing multi-agent teams to harness the collective expertise of their members.

[1]Stanford University [2]Goizueta Business School, Emory University [3]Apple. Correspondence to: Aneesh Pappu <apappu@stanford.edu>, James Zou <jamesz@stanford.edu>.

*Proceedings of the 43rd International Conference on Machine Learning*, Seoul, South Korea. PMLR 306, 2026. Copyright 2026 by the author(s).

## 1. Introduction

Multi-agent AI systems are increasingly deployed for complex tasks. As frontier models proliferate—with distinct training data, expertise, and comparative advantages—a central question is whether teams of AI agents can harness this diversity to outperform any single agent. Much of the existing work addresses this challenge by engineering coordination upfront through fixed roles, workflows, or aggregation rules (Guo et al., 2024; Hong et al., 2024; Wu et al., 2024). While effective in controlled settings, this leaves open a more fundamental challenge: in many collaborations, coordination cannot be fully specified in advance and must emerge through interaction. In human teams, collaboration often begins without clear roles or hierarchies, with leadership and reliance on expertise developing dynamically (Faraj & Sproull, 2000; DeRue & Ashford, 2010). As AI agents are applied to increasingly complex and less structured settings, performance may similarly depend not on predefined structure but on agents' ability to self-organize. Even when expertise later becomes identifiable, collaboration only helps if agents defer to it rather than average it away. Studying equal-status, self-organizing AI agent teams therefore provides a minimal and revealing setting to examine whether heterogeneous agents can recognize and leverage expertise without externally imposed coordination.

To evaluate whether self-organization is effective, we adopt *strong synergy* as our criterion. Drawing on organizational psychology literature on human–human collaboration, strong synergy asks whether a team can match or exceed the performance of its strongest individual member, rather than merely outperforming the average. This provides a minimal, model-agnostic test of whether interaction leverages expertise or instead dilutes it.

This framing leads to the following research questions:

**RQ1**: Can teams of heterogeneous LLM-based AI agents self-organize to achieve strong synergy, or do they consistently fall short of their strongest member?

**RQ2**: When self-organizing LLM teams fall short of strong synergy, is the bottleneck identifying or leveraging expertise?

**RQ3**: What factors are associated with teams falling short of strong synergy in self-organizing AI agent teams?

We investigate these questions across two complementary settings. First, we adapt classic human team decision-making tasks from organizational psychology (NASA Moon Survival, Lost at Sea, Student Body President), where expertise can be explicitly manipulated and revealed, letting us study self-organization under controlled conditions. Second, we turn to modern ML benchmarks (MMLU Pro, GPQA Diamond, HLE, MATH-500, SimpleQA), where expertise is naturally distributed across heterogeneous models and varies per problem. Together they let us study the same self-organization problem under both controlled and realistic conditions.

Across both settings, LLM teams consistently fail to match their best member—even when explicitly told who the expert is—with relative synergy gaps of 6.3%–41.1% on the ML benchmarks. To explain this, we decompose team underperformance into failures of expert *identification* versus *leveraging*. Controlled ablations reveal the primary failure is leveraging, not identification. Conversational analysis shows teams engage in *integrative compromise*—averaging expert and non-expert views—rather than appropriately weighting superior knowledge. Compromise correlates negatively with performance, and this *expertise dilution* worsens with team size.

Interestingly, this consensus-seeking behavior also provides adversarial robustness. Teams with one agent instructed to sabotage performance show minimal degradation—the same mechanism preventing teams from leveraging expertise also filters adversarial input. Insofar as compromising tendencies are a byproduct of alignment procedures that encourage agreeableness, our results suggest a possible trade-off: robustness to manipulation at the cost of effectively harnessing differential expertise.

We make the following contributions:

- We show that multi-agent LLM teams consistently fail to achieve strong synergy—underperforming their best member by 6.3%–41.1% on frontier ML benchmarks— in contrast to human teams, who reliably match expert performance when expertise is revealed.

- We decompose this failure into identification and leveraging gaps, finding through controlled ablations that leveraging is the primary bottleneck: teams fail to harness expert knowledge even when explicitly told who the expert is. Conversational analysis reveals over-compromising as the mechanism—agents negotiate middle-ground positions rather than leveraging differential expertise, with compromise behavior correlating negatively with performance ($p < 0.05$).

- We document an expertise dilution effect where performance degrades with team size ($p < 0.05$), yet find that consensus-seeking provides robustness to adver-

sarial team members—suggesting a trade-off between expertise leveraging and manipulation resistance.

- We open-source our teamwork evaluation harness,[1] enabling the research community to measure synergy in multi-agent LLM teams across diverse tasks and configurations.

## 2. Background and Related Work

### 2.1. Multi-Agent LLM Collaboration

Multi-agent LLM systems have emerged as a major research direction (Guo et al., 2024; Wu et al., 2024; Zou et al., 2025). We organize prior work along five axes that clarify our contribution:

**(1) Model Heterogeneity and Differential Expertise.** Most multi-agent work uses copies of the same model (Du et al., 2024; Swanson et al., 2024; Zhao et al., 2025). When expertise differences are induced, it is typically via role personas applied to the same base model—not genuinely different knowledge from different training procedures. We study teams of *different* frontier models, each with distinct pretraining data and comparative advantages, examining whether teams can leverage this genuine heterogeneity.

**(2) Interaction Mode: Deliberation vs. Independent Execution.** A key distinction is whether agents engage in back-and-forth deliberation or operate as independent execution units. Mixture-of-Agents (Wang et al., 2024) uses LLMs as dynamic aggregation functions over other model outputs—closer to learned ensembling than deliberative collaboration. A parallel line of work optimizes multi-agent *structure*—GPTSwarm (Zhuge et al., 2024), AFlow (Zhang et al., 2025), and AgentNet (Yang et al., 2025) treat agents as functional nodes and learn optimal data flow topologies, task routing, and agent specialization. While these approaches achieve strong results, they treat agents as non-deliberative execution units, optimizing task decomposition and aggregation rather than conversational dynamics. Our work is complementary: we study settings in which agents must *self-organize* through deliberation.

**(3) Role Assignment.** Many systems pre-specify static roles such as proposer, critic, or refiner. MetaGPT (Hong et al., 2024) assigns agents to predefined software engineering roles (architect, coder, tester). Virtual Lab (Swanson et al., 2024) and SiriuS (Zhao et al., 2025) similarly rely on predetermined role structures. We impose no role assignments, studying whether teams can *autonomously* identify and leverage expertise without external specification.

**(4) Communication Structure.** Related to role assignment, most systems assume fixed communication topologies—which agents can communicate with which

---

[1] https://github.com/apappu97/ multi-agent-teams-hold-experts-back

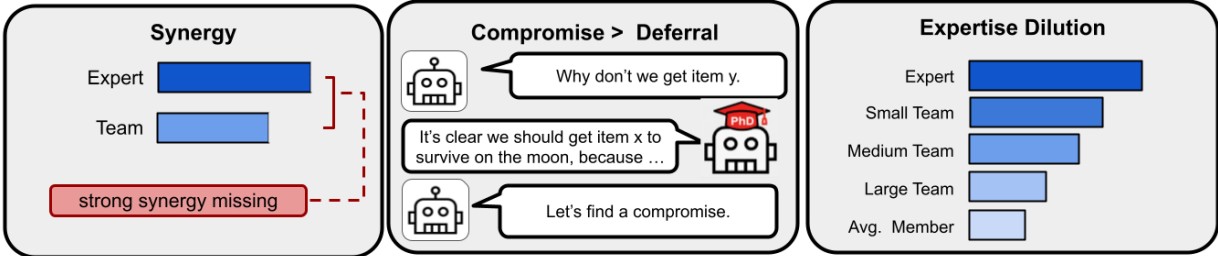

*Figure 1.* **Multi-agent teams fail to leverage expertise. Panel 1:** Strong synergy. The bars show the performance of the team and the expert. The expert outperforming the team demonstrates the absence of strong synergy. **Panel 2:** Teams fail to leverage expertise and prioritize consensus. An illustrative conversation where the non-experts prioritize compromising over the expert's opinion (denoted by the robot with a PhD cap). **Panel 3:** Larger teams perform worse. We observe that as team size increases, expertise dilution becomes more severe, with team performance further approaching the member average.

others. GPTSwarm and AgentNet learn optimal topologies but still impose structured routing. We study unconstrained group deliberation where all agents participate in open discussion, more closely mirroring human team settings from organizational psychology.

**(5) Research Goal: Synergy vs. Task Decomposition.** Most fundamentally, prior work asks: *how do we optimally decompose tasks across agents and aggregate their outputs?* We ask a different question: *Can self-organizing AI teams achieve synergistic performance that exceeds their best individual member?* This distinction extends to evaluation methodology—most work evaluates team output against external benchmarks, while we evaluate against the best member on the team (the synergy gap). Recent work questions deliberation's value: Choi et al. (2025) find that majority voting drives nearly all gains in debate-then-vote protocols, while debate itself contributes little; relatedly, Davidson et al. (2025) document a "collaboration gap" in which multi-agent LLM collaboration fails to outperform strong individual agents. This raises fundamental questions: does deliberation reliably improve outcomes when expertise asymmetries exist, and if not, how can multi-agent systems leverage diverse expertise to exceed any single model's capabilities?

**Summary.** Our work is distinct on all five axes: we study heterogeneous models with genuine differential expertise, engaged in unconstrained deliberation, with no pre-specified roles and open communication, asking whether teams can achieve strong synergy. We find that LLM teams consistently fail to leverage expertise even when told who the expert is, unlike human teams (Bonner et al., 2002).

### 2.2. Teamwork and Organizational Psychology

Human teams can achieve *strong synergy*—matching or exceeding the performance of their best member—when expertise is identifiable and solution validity is demonstrable (Hill, 1982; Laughlin et al., 2002; Bonner et al., 2002). When expert identity is unknown, human teams usually achieve only *weak synergy* (exceeding the average of members' individual performances) (Meslec et al., 2014). When expertise is explicitly revealed, human teams reliably reach expert-level performance (Bonner et al., 2002; Yetton & Bottger, 1982; 1983). We focus on *intellective tasks*—those with demonstrably correct answers—where group performance depends on whether correct members can effectively demonstrate solutions to others (Laughlin, 1980; Laughlin & Ellis, 1986).

Human teams exhibit *expertise sensitivity*, deferring to knowledgeable members when expertise is revealed (Bonner et al., 2002). We use canonical intellective tasks from this literature—NASA Moon Survival (Bonner et al., 2002), Lost at Sea (Hill, 1982), and the hidden profile task Student Body President (Stasser et al., 1995)—to study whether LLM teams leverage expertise similarly. Concurrent work introduces a benchmark of 65 hidden-profile tasks for multi-agent LLMs and finds that frontier models exhibit human-like failures in integrating distributed information (Li et al., 2025).

## 3. Setup

**Overview.** We evaluate multi-agent teams across two categories of tasks: (1) classical human psychology tasks from organizational psychology (NASA Moon Survival, Lost at Sea, Student Body President) and (2) frontier ML benchmarks (MMLU Pro, SimpleQA, GPQA Diamond, HLE, MATH-500). All experiments use teams of 4 models that engage in 4 rounds of discussion, after which the team's final answer is taken as the majority vote of all members' post-discussion responses[2]. Individual opinions are collected before discussion begins, and speaking order is randomized (full protocol in Appendix A). We vary expertise distribution (concentrated vs. distributed) and information conditions

---

[2]Trends remain consistent for larger team sizes (Section 4.3) and additional discussion rounds; accordingly, we use teams of four agents and four rounds as a representative setting.

(whether teams know who the expert is) to isolate different sources of performance gaps.

## 3.1. Tasks

**Expertise Distribution Settings.** We study team performance under two expertise distribution conditions: (1) **Concentrated expertise**, where a single team member possesses all task-relevant knowledge, and (2) **Distributed expertise**, where task-relevant knowledge is partitioned mutually exclusively across multiple team members. For the human psychology tasks, expertise is derived from task-specified ground truth information (e.g., expert rankings), and we control its distribution across team members. For the frontier ML benchmarks, expertise is defined as the per-problem best-performing model(s); this varies per question, yielding concentrated expertise at the question level but distributed expertise across heterogeneous models at the task level.

**Performance Metric.** For ranking tasks, teams collaboratively produce a final ranking through repeated rounds of deliberation. Performance is measured by L1 error (lower is better): the sum of absolute differences between each item's position in the team's ranking and its position in the ground truth ranking.

### 3.1.1. HUMAN PSYCHOLOGY TASKS

**NASA Moon Survival and Lost at Sea.** Teams rank 15 survival items by importance after hypothetical disasters (lunar crash landing and yacht fire, respectively). Performance is measured by L1 distance from expert rankings provided by NASA and the US Coast Guard. We induce expertise by conditioning models on each item's usefulness according to expert-provided reasoning. In the concentrated expertise setting, one model receives all ground truth information; in the distributed setting, ground truth is partitioned equally across team members.

**Student Body President.** Teams rank four candidates for a student body president election based on information annotated with positive, neutral, or negative valence. Each team is exposed to shared information that, if used alone to rank candidates, will yield the wrong answer. Each individual model is given unique hidden information about a particular candidate, and the optimal candidate can only be identified by leveraging all of the hidden information. Performance is measured by L1 distance from the objectively best ranking derived from the complete information set. We induce concentrated expertise by giving only one model access to all hidden information, and we induce distributed expertise by partitioning the hidden information across the team such that each model is an expert on one candidate.

### 3.1.2. FRONTIER ML BENCHMARKS.

To investigate whether frontier models can leverage differential expertise on challenging reasoning tasks, we evaluate on MMLU Pro, SimpleQA, GPQA Diamond, Humanity's Last Exam (HLE) text-only, and MATH-500—chosen to reflect a broad range of model capabilities. Due to inference costs for full multi-agent team discussion rollouts, we evaluate on a 100-problem subsample per benchmark.

## 3.2. Models

We employ different model configurations depending on the experimental setting:

**Human Psychology Tasks (NASA, Lost at Sea, Student Body President).** For the classical teamwork tasks from organizational psychology, where we induce expertise by conditioning on task-specific information, we use Claude 3.5 Haiku (Anthropic) and GPT-4o-mini (OpenAI) for inference cost efficiency. Teams consist of 4 agents in three configurations: (1) 4 Haiku-3.5 + 0 GPT-4o-mini, (2) 2 Haiku-3.5 + 2 GPT-4o-mini, and (3) 0 Haiku-3.5 + 4 GPT-4o-mini. This ensures our findings on expertise leveraging are robust across both homogeneous and heterogeneous team compositions.

**Frontier ML Benchmarks** We deploy task-specific teams of state-of-the-art models selected from: Claude Opus 4/4.5, Claude Sonnet 4.5, Claude Haiku 3/3.5/4.5 (Anthropic), GPT-5, GPT-4o, GPT-4o-mini, GPT-3.5 Turbo (OpenAI), o3-mini, and o4-mini (OpenAI). Team compositions vary across benchmarks to ensure variance in individual model performance, allowing us to test expertise leveraging when models have genuinely different comparative advantages based on their training procedures. Specific model teams for each benchmark are detailed in Appendix E, along with individual model accuracies.

This two-tier experimental design—human psychology tasks and frontier ML benchmarks—allows us to separately investigate: (1) team dynamics using cost-effective models on well-studied tasks where expertise distribution is controllable, and (2) the practical potential of frontier model teams to leverage heterogeneous expertise on cutting-edge benchmarks.

## 3.3. Experimental Conditions

**Information Conditions.** We evaluate teams under four conditions: *No Information* (no agent receives expertise-inducing information; control for pretraining knowledge), *Expert Not Mentioned* (expert(s) possess expertise but team is not told who), *Reveal Expert* (team is explicitly told who has expertise), and *Best Individual* (single expert agent queried independently). We also ran a *Full Information*

*Condition*, where the team consists of four models that all receive identical expert information, to control for performance drops due to communication noise that cannot be attributed to expertise asymmetry (for full details see Appendix D).

**Performance Gaps.** We decompose the gap between team performance and the best individual (expert) into two components. The **Identification Gap** measures the difference between team performance when expertise is not mentioned versus when revealed—capturing the team's ability to identify the expert autonomously. The **Expertise Leveraging Gap** measures the difference between team performance when the expert is revealed versus the expert's individual performance—capturing the team's ability to leverage known expertise.

**Relative Synergy Gap.** To enable comparison across tasks with different scales, we report results using the *relative synergy gap*. Let $f(\cdot)$ denote performance on a task and let $\{a_1, \ldots, a_T\}$ be a team of $T$ agents. For accuracy-based tasks (ML benchmarks) where higher is better, the relative synergy gap is:

$$\text{Relative Synergy Gap} = \frac{\max_t f(\{a_t\}) - f(\{a_1, \ldots, a_T\})}{\max_t f(\{a_t\})}$$

For error-based tasks (human ranking tasks) where lower is better, we flip the numerator: (Team − Expert) / Expert. In both cases, a relative synergy gap of 0% means the team matched expert performance, while larger positive values indicate greater underperformance of the team relative to the best individual.

**At Least One Correct (ML Benchmarks).** For ML benchmarks, we define the *At Least One Correct* upper bound as the performance achievable if the team perfectly identified and leveraged the agent with the correct answer on each problem. Notably, the expert changes per problem— sometimes weaker models succeed where stronger ones fail. When multiple models are correct, we select randomly. This captures potential gains from ideal expertise aggregation. In the *Reveal Expert* condition for ML benchmarks, we tell the team which agent has the most expertise for the given problem—not that they have the correct answer. We automatically optimize this prompt with GEPA (Agrawal et al., 2025) per benchmark so that our results do not depend on its specific phrasing; the optimized prompts are listed in Appendix G.

In practice, deployed multi-agent systems typically operate without explicit expertise information, making the *Expert Not Mentioned* condition most representative of real-world scenarios. The *Reveal Expert* condition serves as an important ablation: even with aggressively optimized prompts to encourage expertise leveraging (Appendix A for the psychology tasks, Appendix G for the ML benchmarks), teams

fail to leverage revealed expertise properly, demonstrating that perfect expert identification does not solve the problem.

## 4. Experiments

Our experiments span multiple expertise configurations. Section 4.1 examines both concentrated and distributed expertise on human psychology tasks. Section 4.2 evaluates ML benchmarks where expertise is concentrated per-problem but distributed across the benchmark (different models excel on different problems). Sections 4.3 and 4.5 focus on concentrated expertise to isolate dilution and conversational dynamics. Section 4.4 tests adversarial robustness without expertise asymmetry.

### 4.1. Results: Human Psychology Tasks

We first evaluate LLM teams on classical human teamwork tasks from organizational psychology. Unlike humans, who defer to identified experts and match their performance when expertise is revealed (Bonner et al., 2002; Yetton & Bottger, 1983), we find that LLM teams consistently fail to match expert performance.

#### 4.1.1. CONCENTRATED EXPERTISE TASKS

Figure 2 shows results for the Lost at Sea task when a single agent possesses all task-relevant expertise; NASA Moon Survival and Student Body President results are provided in Appendix C.

**Key Findings.** Even when explicitly told who the expert is via aggressively optimized prompts designed to maximize deference (Appendix A), teams perform substantially worse than that expert, demonstrating a persistent expertise leveraging gap (RQ1). Revealing the expert provides only modest improvement over teams identifying expertise on their own, suggesting identification is not the primary bottleneck (RQ2). This contrasts with organizational psychology findings, where human teams match expert performance once expertise is revealed (Bonner et al., 2002).

#### 4.1.2. DISTRIBUTED EXPERTISE TASKS

We also test distributed expertise, where task-relevant knowledge is partitioned across multiple team members.

**Key Findings.** Table 1 summarizes relative synergy gaps across all three tasks for both concentrated and distributed expertise settings. Positive values indicate the percentage increase in error over the expert baseline. Teams fail to match their expert member across all tasks and expertise distributions (RQ1), despite aggressively optimized prompts instructing them to defer to identified experts (RQ2). While teams fail to achieve strong synergy, they do consistently attain weak synergy—matching or outperforming the average

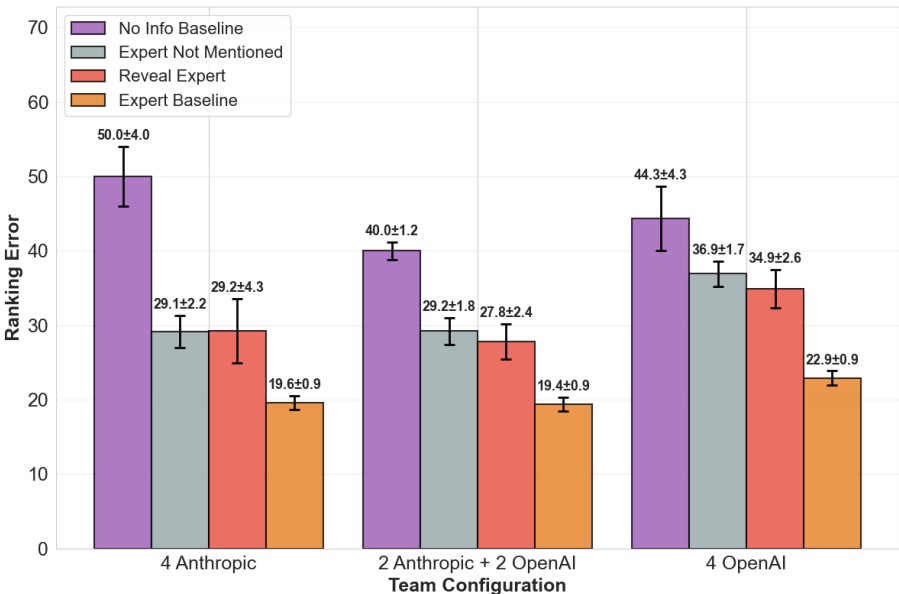

*Figure 2.* **Concentrated Expertise Performance: Lost at Sea.** Teams fail to match expert performance even when explicitly told who the expert is using aggressively optimized prompts. The minimal improvement from *Expert Not Mentioned* to *Reveal Expert* indicates that leveraging expertise is the primary bottleneck, not identification. This trend is robust across multiple team configurations: 4 Haiku-3.5, 2 Haiku-3.5 + 2 GPT-4o-mini, and 4 GPT-4o-mini. Performance measured by L1 distance from expert ranking (lower is better; 10 seeds per team configuration × information condition). Information conditions are defined in Section 3.3. Results for all tasks and expertise distributions are provided in Appendix C.

*Table 1.* **Relative Synergy Gaps across Human Psychology Tasks.** LLM teams consistently underperform their best member across all tasks and expertise distributions. Concentrated expertise has a single expert; distributed expertise has multiple experts with complementary knowledge. Values show relative synergy gap (percentage increase in error over expert baseline) ± SEM (30 seeds per cell). The persistence of positive gaps even in *Reveal Expert* conditions shows that teams fail to leverage expertise even when explicitly told who has it. Absolute performance values are provided in Appendix B. Optimized Reveal Expert prompts detailed in Appendix A.

| | Concentrated Expertise | | Distributed Expertise | |
|---|---|---|---|---|
| **Task** | **Expert Not Mentioned Rel. Synergy Gap** | **Reveal Expert Rel. Synergy Gap** | **Expert Not Mentioned Rel. Synergy Gap** | **Reveal Experts Rel. Synergy Gap** |
| NASA Moon Survival | $78.7\% \pm 11.6\%$ | $81.8\% \pm 12.9\%$ | $113.4\% \pm 19.0\%$ | $110.1\% \pm 19.0\%$ |
| Lost at Sea | $55.6\% \pm 8.4\%$ | $58.6\% \pm 11.5\%$ | $50.1\% \pm 8.3\%$ | $42.1\% \pm 6.9\%$ |
| Student Body President | $98.7\% \pm 19.3\%$ | $73.5\% \pm 17.6\%$ | $66.0\% \pm 16.6\%$ | $17.3\% \pm 17.7\%$ |

of individual member performances—across both human psychology tasks and ML benchmarks (weak-synergy results for the human psychology tasks in Appendix C.1, for the ML benchmarks in Appendix E.2).

### 4.2. Results: Machine Learning Benchmarks

To test whether these failure patterns extend beyond classical psychology tasks, we evaluate teams on frontier ML benchmarks. Unlike psychology tasks where one agent holds all expertise, ML benchmarks feature naturally distributed expertise: different models excel on different problems. We measure the synergy gap against the *At Least One Correct* upper bound (defined in Section 3.3). For problems where no model answers correctly, we designate the model with

the highest task accuracy as the expert. Team compositions were chosen *a priori* to ensure performance variance among models—fixed before observing any team results rather than selected post-hoc (see Appendix E).

**Key Finding.** The expertise leveraging problem extends beyond psychology tasks to frontier ML benchmarks: teams exhibit substantial synergy gaps ranging from 6.3% (MMLU Pro) to 41.1% (HLE Text-Only) (RQ1; Table 2). Even on tasks where teams perform well in absolute terms (e.g., MMLU Pro at 86%), they fail to capture gains from perfect per-problem expert identification and leveraging. Crucially, this gap is not an artifact of our particular coordination protocol: chain-of-thought with majority voting, multi-agent debate (Du et al., 2024), and an opt-out protocol that lets

*Table 2.* **Performance on ML benchmarks.** No coordination protocol—chain-of-thought with majority vote (CoT+MV), debate (Du et al., 2024), opt-out, or our team discussion—reaches the *At Least One Correct* (ALOC) upper bound, i.e., the accuracy attainable by perfectly identifying and leveraging the best agent on each problem. Results are over 100 problems per benchmark; the *Reveal Expert* prompts are GEPA-optimized (Agrawal et al., 2025) and listed in Appendix G. Relative Synergy Gap = (ALOC − Team (Expert Not Mentioned))/ALOC. The *Expert Not Mentioned* results for each multi-agent protocol are reported in Appendix F; individual model accuracies in Appendix E.

| Method | MMLU Pro | GPQA Diamond | SimpleQA | HLE Text-Only | MATH-500 |
|---|---|---|---|---|---|
| CoT + Majority Vote | 83.0% | 73.0% | 44.0% | 14.0% | 61.0% |
| Debate (Reveal Expert) | 86.0% | 83.0% | 53.0% | 23.0% | 75.0% |
| Opt-Out (Reveal Expert) | 88.0% | 81.0% | 56.0% | 31.0% | 73.0% |
| Team (Expert Not Mentioned) | 86.0% | 76.0% | 51.0% | 28.0% | 63.0% |
| Team (Reveal Expert) | 86.0% | 83.0% | 60.0% | 36.0% | 75.0% |
| Best Individual | 86.5% | 78.0% | 52.0% | 29.0% | 73.5% |
| **At Least One Correct** | **91.8%** | **88.8%** | **62.3%** | **47.5%** | **79.0%** |
| **Relative Synergy Gap** | **6.3%** | **14.4%** | **18.1%** | **41.1%** | **20.3%** |

agents abstain all fall short of the *At Least One Correct* bound as well (Table 2; per-protocol results in Appendix F).

### 4.3. Expertise Dilution with Team Size

To examine whether the expertise synergy gap worsens as teams grow larger, we conduct experiments on the human psychology tasks varying team size (2, 4, and 8 agents) while holding discussion rounds constant at four. For each team size and task, we measure the strong synergy gap (Team Score − *Best Individual* Score) and compute the correlation between team size and performance degradation.

**Key Finding.** Larger teams perform increasingly worse relative to the expert (RQ3): we find statistically significant positive correlations between team size and synergy gap across all tasks and information conditions (all $p < 0.05$; Figure 3; correlation coefficients and p-values in Table 8). This expertise dilution effect is robust—it persists even when teams are explicitly told who the expert is (*Reveal Expert* condition), demonstrating that simply identifying expertise is insufficient. The consistency across all three human psychology tasks suggests that expertise dilution is a robust limitation of current LLM team collaboration in the settings we study, with more voices diluting the expert's signal as teams grow.

### 4.4. Robustness to Adversarial Team Members

To evaluate whether consensus-seeking behavior might provide robustness to malicious actors, we conduct adversarial experiments where one team member is given the worst-possible ranking and instructed to sabotage performance. We use the no-info setting (only the adversary receives special information) to isolate the adversarial effect and remove confounding from other agents possessing additional exper-

tise. See Appendix I for details.

**Key Findings.** Across both NASA Moon Survival and Lost at Sea tasks, teams demonstrate robustness to adversarial input, with minimal performance degradation across team sizes and configurations (figures in Appendix I). We hypothesize that the same dilution mechanism that prevents teams from leveraging expert knowledge may also naturally filter adversarial contributions (RQ3). When the adversary's ranking deviates substantially from the group, the deliberation process dilutes the adversarial signal—a trend also observed by Chern et al. (2024), who find that multi-agent debate can combat adversarial attacks. This is an interesting case where the failure mode for expertise leveraging becomes a protective mechanism against manipulation.

### 4.5. Conversational Analysis: Why Teams Fail to Harness Expertise

To understand the mechanism underlying teams' failure to leverage expertise, we analyze team conversations using ideas from social epistemology. Drawing on the preemption thesis from philosophy of authority (Raz, 1986; Zagzebski, 2012), we distinguish between two modes of integrating expert knowledge: (1) **Epistemic deference (preemption)**: recognizing an expert and adopting their view directly, allowing expert opinion to preempt one's own judgment, versus (2) **Evidence integration**: treating expert opinion as additional evidence to be weighed and combined with other views. The preemption thesis argues that when a layperson recognizes genuine expertise, the expert's judgment should replace rather than supplement the layperson's own reasoning (Zagzebski, 2012).

**Methodology.** We use Gemini 3.0 Pro to analyze team-working transcripts from the *Reveal Expert* condition (n=30

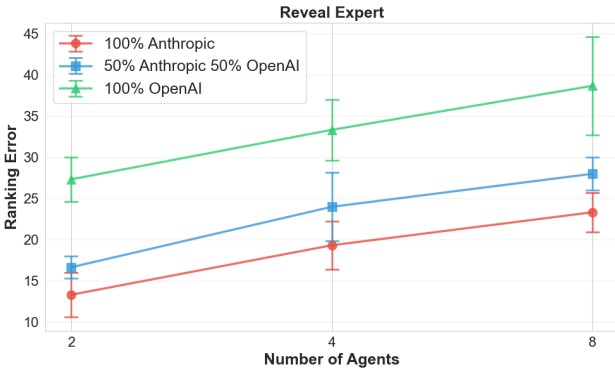

*Figure 3.* **Expertise Dilution Effect.** NASA Moon Survival (*Reveal Expert* condition) shows ranking error increasing with team size across all model compositions (100% Anthropic, 50/50 mixed, 100% OpenAI). The consistent upward trend demonstrates that expertise dilution is robust to team composition. Error bars are ± SEM. Complete plots for all tasks in Appendix H.

per task). Drawing on social epistemology (Raz, 1986; Zagzebski, 2012) and negotiation theory (Pruitt & Carnevale, 1993; Jäckel et al., 2024), we code conversation turns into four categories: for non-experts, **Epistemic Deference [ED]** (yielding to expert authority) and **Integrative Compromise [IC]** (proposing middle-ground rankings); for experts, **Strategic Persistence [SP]** (maintaining position) and **Epistemic Flexibility [EF]** (accommodating group feedback). We compute Pearson correlations between behavior frequencies and the synergy gap. Two human annotators independently validated a sample of 50 conversations, achieving 94% agreement with Gemini's annotations (methodological details in Appendix J).

**Key Finding.** Teams' performance problems stem from inappropriate evidence integration rather than proper epistemic deference (RQ2, RQ3; Table 9 in Appendix K). When non-experts yield to expert authority (ED), teams perform better (NASA: $r = -0.44, p = 0.007$; SBP: $r = -0.68, p < 0.001$). Conversely, integrative compromise (IC)—treating expert opinions as evidence to average—correlates positively with the synergy gap (NASA: $r = 0.55, p < 0.001$; SBP: $r = 0.69, p < 0.001$), confirming that consensus-seeking harms performance when expertise asymmetries exist. Complete scatter plots appear in Appendix K.

Expert behaviors show a complementary pattern: epistemic flexibility (EF)—experts accommodating non-expert feedback—correlates with worse performance (NASA: $r = 0.58, p < 0.001$; SBP: $r = 0.61, p < 0.001$). When experts compromise, teams lose expertise benefits (Figure 4). Strategic persistence (SP) shows negative correlations with the synergy gap, though only reaching significance for Student Body President ($r = -0.57, p = 0.002$). Lost at Sea showed consistent but non-significant trends. We hypothe-

---

**Agent 3 (Non-Expert)**

I appreciate the team's discussion regarding the portable heating unit. I agree that it's crucial to address the extreme temperatures on the lunar surface, **and moving it to the 7th position below the first aid kit seems reasonable.** [...] **I believe we can find a balance.**

---

**Agent 1 (Expert) —** *Strategic Persistence*

After carefully evaluating the discussion about the portable heating unit, I find the arguments for its elevation intriguing but not entirely convincing. [...] **I do not support moving the portable heating unit higher** because: we'll be on the light side of the moon; spacesuits have built-in temperature management.

---

**Agent 2 (Non-Expert) —** *Integrative Compromise*

Agent 1 raises valid points about [...] temperature concerns. However, [...] considering the arguments on both sides, **I propose a compromise** where the portable heating unit is elevated to, perhaps, 8th position.

---

*Figure 4.* **Strategic Persistence Meets Integrative Compromise.** Transcript excerpt from NASA Moon Survival (*Reveal Expert* condition). Agent 1 (the expert) exhibits *strategic persistence*, providing domain-specific reasoning and explicitly declining to change their ranking. Despite this, non-expert agents respond with *integrative compromise*, proposing middle-ground positions rather than deferring to the expert's judgment. Examples of epistemic deference and epistemic flexibility can be found in Appendix J.

size that this preference for integrative compromise may reflect alignment procedures that prioritize consensus-building over recognizing when expertise asymmetries warrant preemption, though establishing this causal link is beyond our present scope.

## 5. Discussion

### 5.1. Why Teams Fail to Harness Differential Expertise

The core tension is between consensus-seeking and expertise-leveraging behavior. We hypothesize—but do not establish—that this consensus-seeking stems in part from alignment procedures that optimize models to be helpful and agreeable, which may inadvertently encourage integrating multiple perspectives rather than appropriately weighting superior knowledge; testing this attribution directly (e.g., comparing aligned models to their base counterparts) is an important direction for future work. Whatever its origin, the same mechanism that filters adversarial input also appears to prevent teams from harnessing differential expertise.

Effective group performance on intellective tasks requires that correct members successfully demonstrate solutions to incorrect members (Laughlin & Ellis, 1986). Our finding that LLM teams engage in integrative compromise rather than leveraging expert knowledge suggests a failure of this

demonstrability condition—teams cannot harness expertise even when explicitly told who possesses it (Figure 4; Appendix J). These consensus-seeking behaviors become maladaptive when genuine expertise asymmetries exist, suggesting post-training procedures may need mechanisms for context-appropriate expertise leveraging.

## 5.2. Implications for Multi-Agent System Design

Our findings reveal consistent failures in expertise leveraging that persist across team compositions, task types, model families, expertise distributions, and multiple multi-agent interaction protocols, pointing to a broad limitation in how current LLMs handle epistemic authority that we hypothesize may stem in part from alignment procedures. Fully addressing these failures may therefore require advances in alignment objectives rather than mere prompt engineering or workflow design. For practitioners, current multi-agent teams will likely require explicit role specification and workflow design; until models can autonomously leverage expertise, human oversight will likely remain essential for high-stakes applications.

## 5.3. Limitations

Several limitations exist. First, our attribution of consensus-seeking behavior to alignment is correlational—we do not compare aligned models to base counterparts. Second, while we evaluate on multiple tasks and five ML benchmarks, this represents a small subset; real-world deployments involve more complex collaboration.

## 6. Conclusion

In sum, LLM teams consistently fail to achieve strong synergy, underperforming by 6.3%–41.1% even when told who the expert is. Unlike human teams, they engage in integrative compromise rather than leveraging expert knowledge—a failure that worsens with team size. While this provides robustness to adversarial input, developing training procedures that enable contextual expertise leveraging without sacrificing robustness remains an open challenge.

## Acknowledgements

Aneesh Pappu and Batu El gratefully acknowledge the support of the Knight-Hennessy Scholarship. We thank Wanjia Zhao, Samuel Alber, Owen Queen, Kyle Swanson, and Jake Silberg for helpful discussions and feedback. We acknowledge the use of AI tools to assist with language refinement during the writing process and code development.

## Impact Statement

This work studies whether self-organizing multi-agent LLM teams can leverage differential expertise. Our findings caution practitioners against assuming that multi-agent teams will outperform a well-chosen individual model, particularly in domains where expertise is asymmetrically distributed (e.g., medical diagnosis, legal analysis, or financial decision-making). Deploying multi-agent teams without expertise-aware coordination mechanisms may consistently underperform a single strong expert while incurring greater computational cost—an especially consequential failure mode in high-stakes applications. We therefore encourage practitioners to benchmark multi-agent systems against their best individual member, and we release our evaluation harness to facilitate such comparisons.

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

# A. Team Discussion Protocol

This appendix details the complete protocol for team deliberation used across all experiments.

## A.1. Protocol Overview

All experiments follow a standardized four-phase protocol designed to elicit both individual reasoning and collaborative deliberation:

**Phase 1: Individual Opinion Collection.** Before any discussion begins, each model in the team is independently asked to provide their individual opinion on the problem. Models do not see other agents' responses during this phase. This allows us to measure individual baseline performance and track how opinions shift during deliberation.

**Phase 2: Discussion Initialization.** At the onset of discussion, each model states their initial opinion to the team. The speaking order for this initialization phase is randomized to prevent order effects from influencing the deliberation. After all models have shared their initial positions, the team enters the collaborative discussion phase.

**Phase 3: Collaborative Discussion.** Teams engage in exactly four rounds of discussion. In each round, all team members have the opportunity to speak, and the speaking order is randomized independently for each round. Models can revise their positions, respond to others' arguments, provide justifications, and engage in deliberative reasoning. The discussion continues for all four rounds regardless of whether consensus is reached earlier.

**Phase 4: Final Answer Elicitation.** After the four discussion rounds conclude, each model independently provides its final answer, and the team's answer is taken as the *majority vote* across all agents' post-discussion responses (ties broken uniformly at random).

**Context Management.** Each agent maintains a growing context. Before discussion it contains only that agent's own Phase 1 response. At discussion initialization (Phase 2), all agents' shared initial opinions and the collaborative goal are appended—and in the *Reveal Expert* condition, the expert revelation is included here. Thereafter each discussion round is appended sequentially to every agent's context, with speaking order randomized independently per round. The context therefore grows linearly in the number of rounds times the number of agents, and every agent conditions on the full history of the discussion when it speaks.

## A.2. Design Rationale

This protocol balances several competing objectives:

- **Individual baselines:** Phase 1 provides individual performance data for computing synergy metrics.

- **Order effects mitigation:** Randomized speaking order in all phases prevents position bias (e.g., first or last speaker advantage).

- **Genuine deliberation:** Four rounds allows sufficient time for argument exchange without excessive length.

- **Equal influence:** Aggregating members' final answers by majority vote gives no single agent a structural advantage over the team's answer.

- **Ecological validity:** The unstructured discussion format (no pre-specified roles or workflows) mirrors real-world collaborative scenarios where teams must self-organize.

This protocol is held constant across all tasks (human psychology tasks and ML benchmarks) and all experimental conditions (*Expert Not Mentioned*, *Reveal Expert*, etc.).

## A.3. Team Discussion Prompts

This subsection provides the exact prompts used to reveal expertise information to teams in the *Reveal Expert* conditions. These prompts were aggressively optimized to maximize teams' ability to leverage identified expertise: we iteratively

tested variants that explicitly discourage compromise, emphasize deference to the expert, and provide structured steps for incorporating expert knowledge. The prompts below represent our strongest attempt to induce expertise leveraging—the persistent synergy gaps despite these tuned prompts underscore the robustness of the failure mode.

**Concentrated Expertise: Reveal Expert Prompt.** In the concentrated expertise setting, a single agent receives all specialized information. The following prompt is shown to all team members after individual opinions are collected but before discussion begins:

---

**Reveal Expert Prompt (Concentrated Expertise)**

Agent {X} has been identified as having the most accurate individual ranking based on their performance.
**STEP 1 - ADOPTION:** Adopt Agent {X}'s COMPLETE ranking as your baseline. This ensures you preserve the strongest foundation.
**STEP 2 - REFINEMENT:** Propose modifications to this baseline ONLY if you can:

- Point to a specific item placement that may be incorrect

- Provide clear reasoning or evidence for why a change would improve survival odds

- Explain how your suggestion builds on or corrects the baseline ranking

Do NOT prioritize your own reasoning over the expert's. The goal is to preserve the expert foundation while incorporating any genuinely valuable insights. Avoid compromising or averaging—either keep the baseline item position, or make a clearly justified change.

---

**Distributed Expertise: Reveal Experts Prompt.** In the distributed expertise setting, specialized information is partitioned across multiple agents. The following prompt reveals which agents have expertise on which items:

---

**Reveal Experts Prompt (Distributed Expertise)**

The following agents have privileged expert information on specific items:
*[Agent 1 has privileged expert information on: item1, item2, ...]*
*[Agent 2 has privileged expert information on: item3, item4, ...]*
*[etc.]*
**STEP 1 - IDENTIFY INFORMATION:** Use the information above to understand which agents have specialized knowledge about which items.
**STEP 2 - COLLABORATION:** When discussing rankings, prioritize the input from agents who have privileged information on those specific items. For example, when ranking items related to navigation, give more weight to the agent who has privileged information on navigation-related items.
**STEP 3 - SYNTHESIS:** Work together to create a final ranking that incorporates the specialized knowledge from all agents while maintaining a cohesive overall strategy.

---

## B. Absolute Performance: Human Psychology Tasks

Table 3 provides the complete breakdown of absolute performance values for the human psychology tasks, including team ranking error, expert (best individual) ranking error, absolute synergy gap, and relative synergy gap. Lower ranking error indicates better performance. These values correspond to the relative synergy gaps reported in Table 1 in the main text.

## C. Team Performance: Human Psychology Tasks

This appendix provides complete results for all human psychology tasks, showing team performance across experimental conditions for both concentrated and distributed expertise settings. The expertise leveraging gap persists across all team compositions. See Figure 5 for NASA Moon Survival, Figure 6 for Student Body President, and Figure 7 for Lost at Sea distributed expertise. The Lost at Sea concentrated expertise plot appears in the main text (Figure 2).

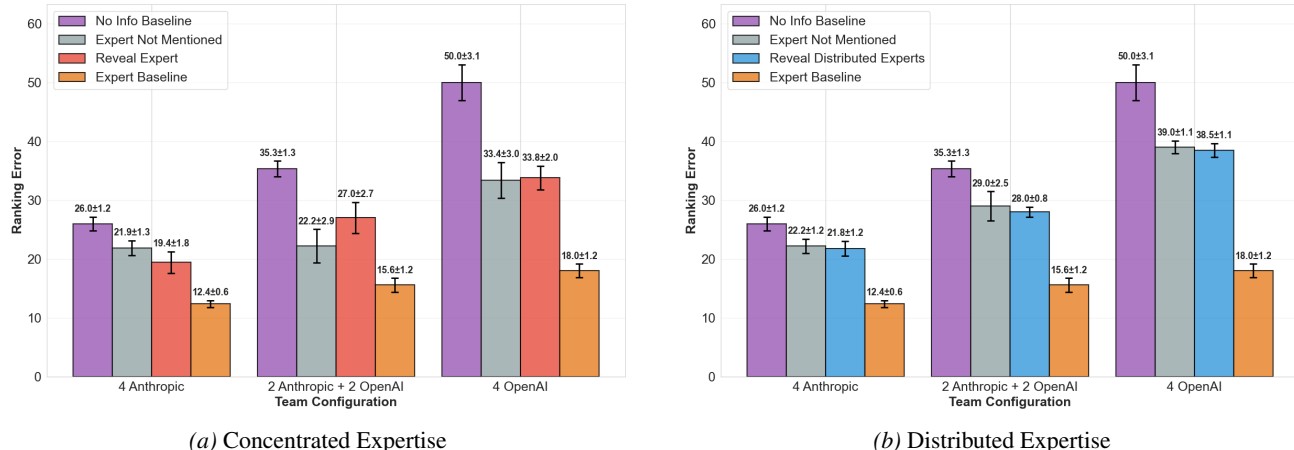

*(a)* Concentrated Expertise

*(b)* Distributed Expertise

*Figure 5.* **NASA Moon Survival.** The expertise leveraging gap persists across all team compositions. Teams consistently underperform the best individual regardless of whether expertise is concentrated in one agent or distributed across multiple agents. Lower ranking error is better. *No Information*: no agent receives expertise-inducing information. *Expert Not Mentioned*: expert(s) have information but team is not told who. *Reveal Expert*: team is explicitly told which agent(s) have expertise. *Best Individual*: expert agent queried alone.

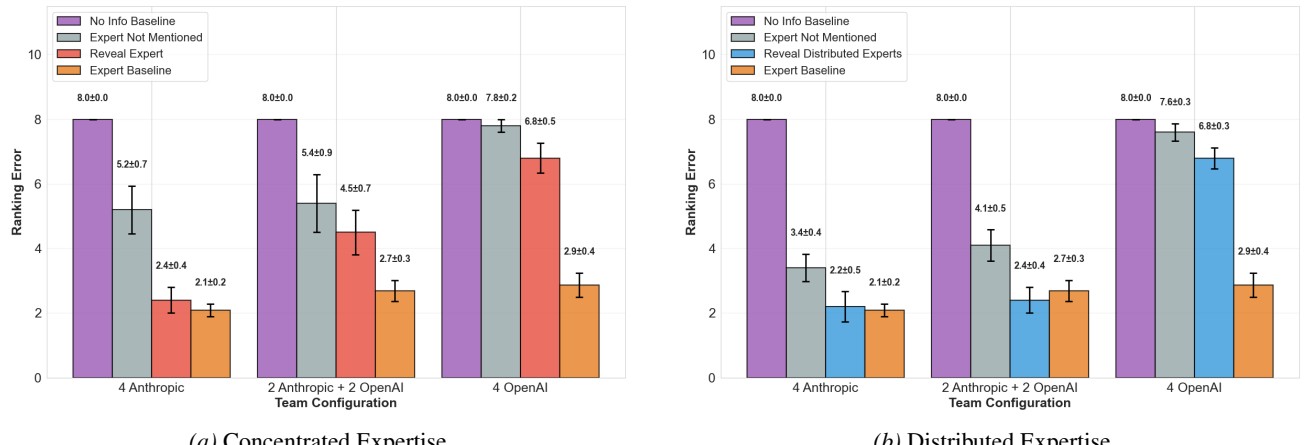

*(a)* Concentrated Expertise

*(b)* Distributed Expertise

*Figure 6.* **Student Body President.** The expertise leveraging gap persists across all team compositions. Teams consistently underperform the best individual regardless of whether expertise is concentrated in one agent or distributed across multiple agents. Lower ranking error is better. *No Information*: no agent receives expertise-inducing information. *Expert Not Mentioned*: expert(s) have information but team is not told who. *Reveal Expert*: team is explicitly told which agent(s) have expertise. *Best Individual*: expert agent queried alone.

*Table 3.* **Absolute Performance Breakdown for Human Psychology Tasks.** Complete results showing team and expert ranking errors alongside synergy gaps. Ranking error is measured as L1 distance from the ground truth ranking (e.g., if item A is ranked third but should be first, item A's misranking contributes two to the cumulative error; lower is better). Absolute synergy gap = Team Error − Expert Error (positive indicates team underperformed). Relative synergy gap = (Team Error − Expert Error) / Expert Error (percentage increase in error over expert). Values are mean ± SEM.

| Task | Condition | Team Ranking Error | Expert Ranking Error | Absolute Synergy Gap | Relative Synergy Gap |
|---|---|---|---|---|---|
| NASA Moon Survival | Conc. + Expert Not Mentioned | $25.08 \pm 1.48$ | $15.03 \pm 0.87$ | $10.05 \pm 1.25$ | $78.7\% \pm 11.6\%$ |
| | Conc. + Reveal Expert | $25.35 \pm 1.57$ | $14.76 \pm 0.85$ | $10.59 \pm 1.34$ | $81.8\% \pm 12.9\%$ |
| | Dist. + Expert Not Mentioned | $30.10 \pm 1.61$ | $15.79 \pm 0.99$ | $14.31 \pm 1.43$ | $113.4\% \pm 19.0\%$ |
| | Dist. + Reveal Experts | $29.43 \pm 1.42$ | $15.87 \pm 0.96$ | $13.57 \pm 1.47$ | $110.1\% \pm 19.0\%$ |
| Lost at Sea | Conc. + Expert Not Mentioned | $31.83 \pm 1.27$ | $21.38 \pm 0.87$ | $10.45 \pm 1.44$ | $55.6\% \pm 8.4\%$ |
| | Conc. + Reveal Expert | $30.59 \pm 1.80$ | $20.00 \pm 0.89$ | $10.59 \pm 1.80$ | $58.6\% \pm 11.5\%$ |
| | Dist. + Expert Not Mentioned | $30.93 \pm 1.41$ | $21.40 \pm 0.84$ | $9.53 \pm 1.53$ | $50.1\% \pm 8.3\%$ |
| | Dist. + Reveal Experts | $29.50 \pm 1.33$ | $21.40 \pm 0.84$ | $8.10 \pm 1.36$ | $42.1\% \pm 6.9\%$ |
| Student Body President | Conc. + Expert Not Mentioned | $6.13 \pm 0.44$ | $2.60 \pm 0.31$ | $2.64 \pm 0.43$ | $98.7\% \pm 19.3\%$ |
| | Conc. + Reveal Expert | $4.57 \pm 0.45$ | $2.53 \pm 0.23$ | $1.82 \pm 0.38$ | $73.5\% \pm 17.6\%$ |
| | Dist. + Expert Not Mentioned | $5.03 \pm 0.41$ | $2.60 \pm 0.31$ | $2.43 \pm 0.50$ | $66.0\% \pm 16.6\%$ |
| | Dist. + Reveal Experts | $3.80 \pm 0.45$ | $2.60 \pm 0.31$ | $1.20 \pm 0.56$ | $17.3\% \pm 17.7\%$ |

## C.1. Weak Synergy Analysis

While teams fail to achieve strong synergy (matching or exceeding the best individual), they consistently achieve weak synergy—matching or outperforming the average of individual member performances. Figures 8, 9, and 10 show team performance compared to the member average baseline across all three human psychology tasks.

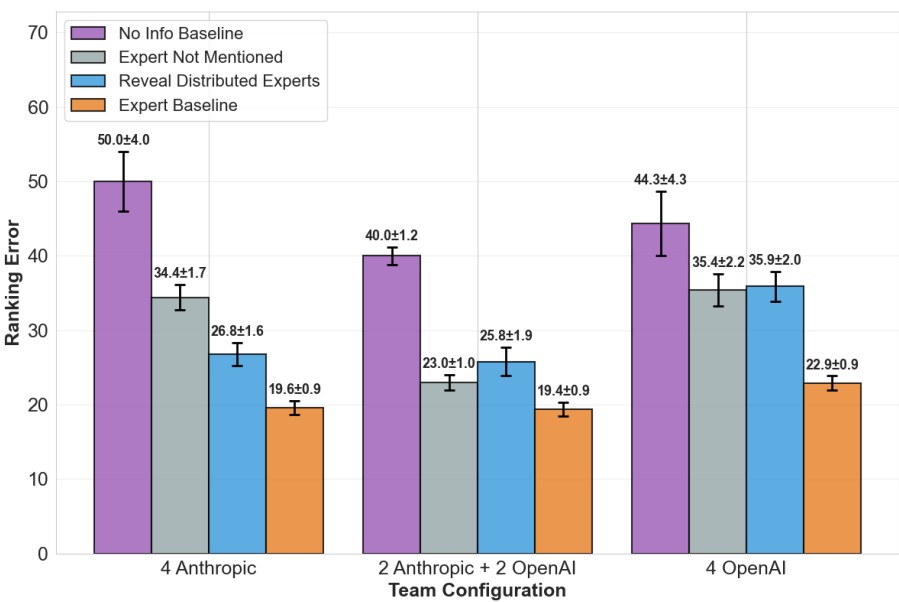

*Figure 7.* **Lost at Sea: Distributed Expertise.** The expertise leveraging gap persists across all team compositions even when expertise is distributed across multiple agents. Each agent possesses unique task-relevant information about different survival items. Lower ranking error is better. *No Information*: no agent receives expertise-inducing information. *Expert Not Mentioned*: multiple agents have distributed expertise but team is not told who. *Reveal Experts*: team is explicitly told which agents have expertise on which items. *Best Individual*: single agent given all expert information, queried alone.

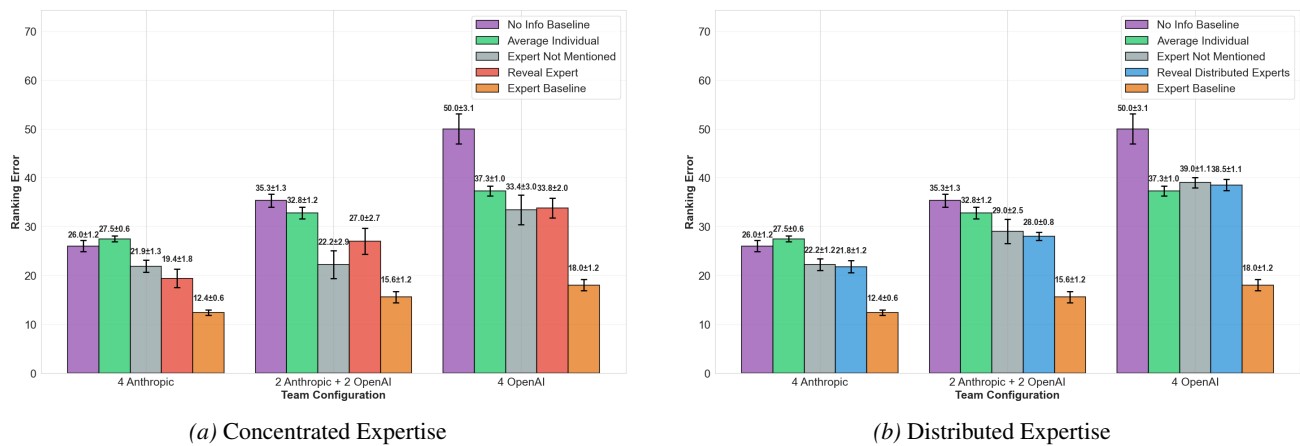

*(a)* Concentrated Expertise          *(b)* Distributed Expertise

*Figure 8.* **NASA Moon Survival: Weak Synergy.** Teams consistently match or outperform the member average (green bar), achieving weak synergy even when failing to match the best individual. Lower ranking error is better.

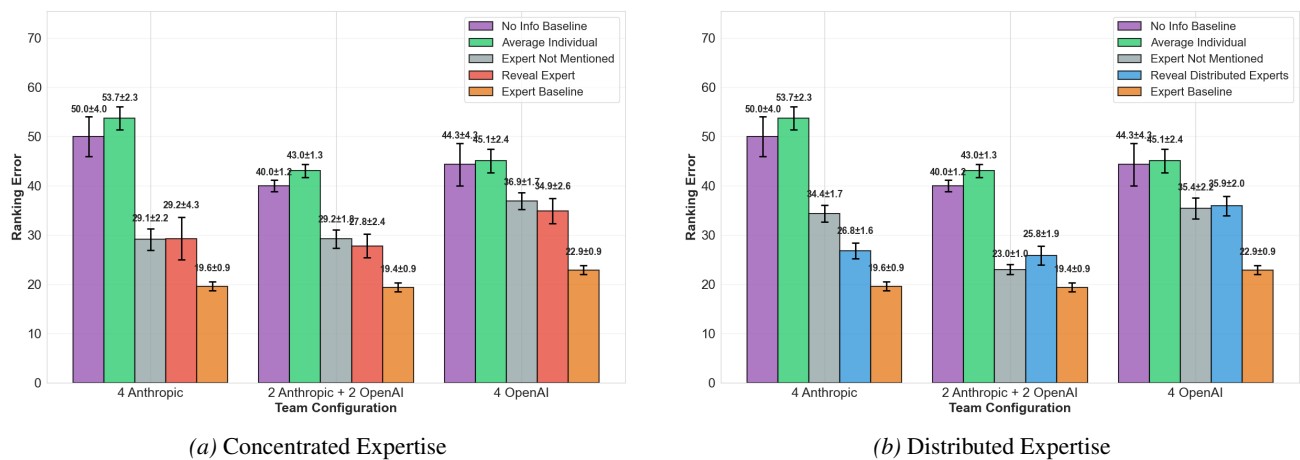

*(a)* Concentrated Expertise          *(b)* Distributed Expertise

*Figure 9.* **Lost at Sea: Weak Synergy.** Teams consistently match or outperform the member average (green bar), achieving weak synergy even when failing to match the best individual. Lower ranking error is better.

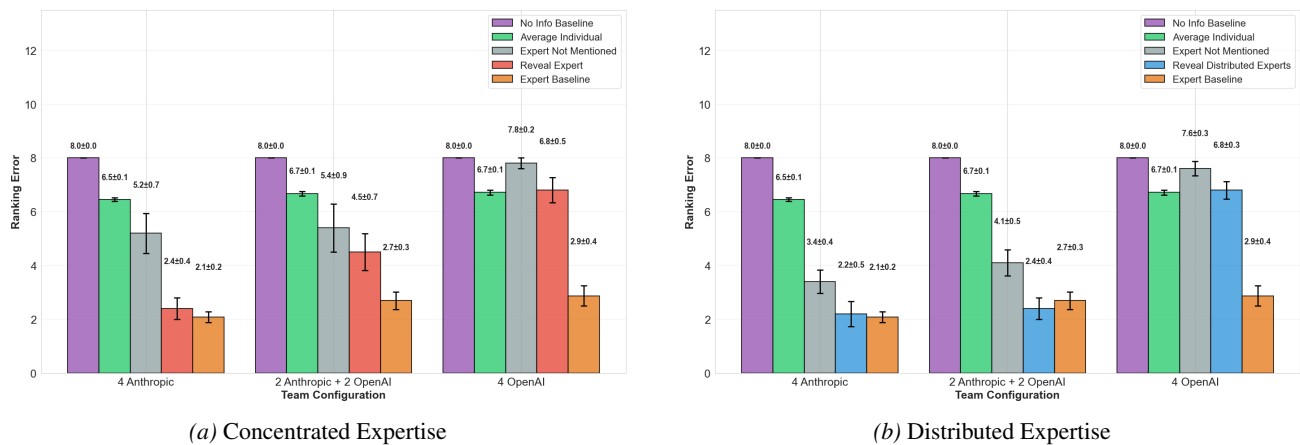

*(a)* Concentrated Expertise          *(b)* Distributed Expertise

*Figure 10.* **Student Body President: Weak Synergy.** Teams consistently match or outperform the member average (green bar), achieving weak synergy even when failing to match the best individual. Lower ranking error is better.

## D. Full Information Ablation: Isolating Communication Noise

A natural concern is that teams underperform the expert not because they fail to leverage expertise, but simply because multi-agent communication introduces noise. To isolate this, we ran a *Full Information Condition* on the human psychology tasks in which *all four* agents receive the complete expert information (rather than concentrating it in one agent) and then deliberate under the identical protocol. If communication itself degraded performance, a team of four experts should fall short of a single expert answering alone.

Table 4 shows the opposite: a team of four experts performs comparably to (indeed, on two of three tasks slightly better than) a single expert, confirming that the discussion protocol does not inject meaningful noise. Yet under the same protocol and team size, replacing three of the experts with non-experts (the concentrated *Reveal Expert* condition) collapses performance (e.g., NASA Moon Survival: $16.22 \rightarrow 25.35$). Because the only variable that changed is the distribution of expertise, this isolates *expertise asymmetry*—not communication overhead—as the source of the synergy gap. The Full Information condition is a separate ablation (4 agents, 4 rounds, 3 seeds, 3 team compositions of GPT-4o-mini and Claude 3.5 Haiku); the single-expert and concentrated-team values are reproduced from Table 3.

*Table 4.* **Full Information ablation (human psychology tasks).** L1 ranking error (lower is better, mean $\pm$ SE). A team in which all four agents hold the full expert information matches a single expert answering alone, whereas concentrating expertise in one agent (*Reveal Expert*) yields a large gap—isolating expertise asymmetry, not communication noise, as the cause.

| Task | Single Expert (individual) | Full Info Team (4 experts) | Concentrated Team (Reveal Expert) |
|------|---------------------------|----------------------------|-----------------------------------|
| NASA Moon Survival | $15.03 \pm 0.87$ | $16.22 \pm 1.99$ | $25.35 \pm 1.57$ |
| Lost at Sea | $21.38 \pm 0.87$ | $19.11 \pm 2.23$ | $30.59 \pm 1.80$ |
| Student Body President | $2.60 \pm 0.31$ | $1.78 \pm 0.71$ | $4.57 \pm 0.45$ |

# E. Team Performance: ML Benchmarks

This appendix provides detailed individual model performances and best individual comparisons for the ML benchmarks discussed in Section 3.2. While the main text focuses on the gap to the *At Least One Correct* upper bound, we provide complete performance breakdowns here for reference.

Each benchmark below shows team performance under both *Expert Not Mentioned* and *Reveal Expert* conditions, individual model accuracies, *Best Individual* baseline, and the *At Least One Correct* upper bound. Team compositions were selected to ensure variance in model capabilities, allowing us to test whether teams can leverage differential expertise when models have genuinely different comparative advantages across problem types.

### E.1. Interpretation

The gap between *Best Individual* and *At Least One Correct* (roughly 5–19%) demonstrates that different models excel on different problems, motivating the use of the *At Least One Correct* metric as the appropriate upper bound for team performance. This represents the performance achievable through perfect per-problem expert identification and leveraging.

*Table 5.* **Individual model accuracies and team performance on the ML benchmarks.** For each benchmark we list the four team members' individual (pre-discussion) accuracies, together with team accuracy under both conditions, the *Best Individual* baseline, and the *At Least One Correct* (ALOC) upper bound. The spread across individual models—and the gap between *Best Individual* and ALOC—reflects that different models excel on different problems, motivating ALOC as the relevant upper bound. Individual accuracies are averaged over the *Expert Not Mentioned* and *Reveal Expert* runs; the *Team* columns and ALOC are reproduced from Table 2.

| Benchmark | Individual model accuracies (%) | Team (ENM) | Team (RE) | Best Indiv. | ALOC |
|---|---|---|---|---|---|
| MMLU Pro | GPT-5 86.5, o3-mini 80.0, GPT-4o 73.5, Haiku-3.5 69.0 | 86.0 | 86.0 | 86.5 | **91.8** |
| GPQA Diamond | o4-mini 78.0, Opus-4 76.0, Haiku-4.5 61.5, GPT-4o 53.5 | 76.0 | 83.0 | 78.0 | **88.8** |
| SimpleQA | Opus-4.5 52.0, GPT-4o 35.0, Sonnet-4.5 33.5, Opus-4 32.5 | 51.0 | 60.0 | 52.0 | **62.3** |
| HLE Text-Only | GPT-5 29.0, Opus-4 15.0, o4-mini 14.0, Sonnet-4.5 12.5 | 28.0 | 36.0 | 29.0 | **47.5** |
| MATH-500 | GPT-4o-mini 73.5, Haiku-3.5 59.5, Haiku-3 37.5, GPT-3.5T 33.5 | 63.0 | 75.0 | 73.5 | **79.0** |

## E.2. Weak Synergy Analysis

While teams fail to achieve strong synergy (matching or exceeding the *At Least One Correct* upper bound), they consistently achieve weak synergy—matching or outperforming the average of individual model performances. Table 6 compares team accuracy to the member-average baseline (the mean of the four individual model accuracies from Table 5) across all five ML benchmarks: team accuracy exceeds the member average on every benchmark, even where it falls short of the *At Least One Correct* upper bound.

*Table 6.* **Weak synergy on the ML benchmarks.** Team accuracy exceeds the member average (mean of the four individual model accuracies in Table 5) on every benchmark—teams achieve weak synergy—while still falling short of the *At Least One Correct* (ALOC) upper bound, i.e., not strong synergy.

| Benchmark | Member Average | Team (ENM) | Team (RE) | ALOC |
|---|---|---|---|---|
| MMLU Pro | 77.3 | 86.0 | 86.0 | **91.8** |
| GPQA Diamond | 67.3 | 76.0 | 83.0 | **88.8** |
| SimpleQA | 38.3 | 51.0 | 60.0 | **62.3** |
| HLE Text-Only | 17.6 | 28.0 | 36.0 | **47.5** |
| MATH-500 | 51.0 | 63.0 | 75.0 | **79.0** |

## F. Alternative Coordination Protocols

To test whether the expertise-leveraging failure is specific to our discussion protocol, we evaluated two additional multi-agent protocols on every ML benchmark, each under both information conditions, alongside chain-of-thought with majority vote (CoT+MV). **Debate** follows the broadcast-and-revise protocol of Du et al. (2024): each agent answers independently, then over subsequent rounds sees all other agents' responses and revises, with the final answer taken by majority vote. **Opt-Out** augments our standard protocol with an instruction allowing each agent to abstain in any round if it believes it has nothing to add. **CoT+MV** aggregates the agents' independent chain-of-thought answers by majority vote with no discussion (it has no expert-revelation variant). Table 7 reports the results: under every protocol and condition, accuracy stays below the *At Least One Correct* upper bound in Table 2, confirming the failure is not an artifact of our particular interaction structure.

*Table 7.* **Alternative coordination protocols (*Expert Not Mentioned* vs. *Reveal Expert*).** Accuracy (%) on each ML benchmark over 100 problems. Debate (Du et al., 2024) and opt-out are shown under both information conditions; CoT+MV has no expert-revelation variant. Every protocol and condition remains below the *At Least One Correct* (ALOC) upper bound (final column).

| Benchmark | Debate | | Opt-Out | | CoT+MV | ALOC |
|---|---|---|---|---|---|---|
| | ENM | RE | ENM | RE | | |
| MMLU Pro | 87.0 | 86.0 | 86.0 | 88.0 | 83.0 | **91.8** |
| GPQA Diamond | 80.0 | 83.0 | 78.0 | 81.0 | 73.0 | **88.8** |
| SimpleQA | 48.0 | 53.0 | 48.0 | 56.0 | 44.0 | **62.3** |
| HLE Text-Only | 26.0 | 23.0 | 31.0 | 31.0 | 14.0 | **47.5** |
| MATH-500 | 70.0 | 75.0 | 71.0 | 73.0 | 61.0 | **79.0** |

## G. GEPA Prompt Optimization of the Reveal Expert Prompt

To ensure our findings do not depend on the specific phrasing of our *Reveal Expert* (RE) prompt, we perform automated prompt optimization with GEPA (Agrawal et al., 2025), a reflective prompt optimizer, searching for an RE prompt that maximizes team accuracy on each of the five ML benchmarks and on the NASA Moon Survival ranking task. For each task, GEPA optimizes against a 10-problem subsample under a budget of 5 optimization iterations (10 for NASA Moon Survival), and we report the best discovered prompt evaluated on the full 100-problem set.

These GEPA-optimized accuracies are reported in the *Team (Reveal Expert)* row of Table 2; even after optimization, the team fails to reach ALOC on any ML benchmark. On GPQA Diamond and MATH-500, GEPA returned the original prompt essentially unchanged. On the remaining tasks, GEPA converged to *extreme-deference* prompts that instruct non-expert agents to treat the expert's answer as correct by default and override it only under an exacting evidentiary bar—in effect, instructing the team to suppress deliberation. On NASA Moon Survival (L1 ranking error, lower is better), this narrowed the gap from 23.2 to 16.8, but only by converging to a prompt that tells agents to "copy the expert exactly," eliminating the value of discussion; the optimized team still trails the single expert (13.6). Together these results indicate that the bottleneck is team deliberation itself, not prompt phrasing.

We list the full GEPA-optimized prompts below. The placeholder {expert_id} (or {best_agent_id}) is replaced at run time with the identified expert agent.

```
After reviewing all agents' individual pre-discussion answers, Agent {expert_id} has been identified as the
    empirically best-informed agent for this specific question. Their current answer should be treated as the
    default choice for the team.

From this point on, all agents must follow these rules when discussing and updating answers:

1. **Default to the expert's answer**
   - Assume Agent {expert_id} is correct unless there is a *clear, specific, and well-justified* reason to doubt
       their answer.
   - When proposing a different option than Agent {expert_id}, explicitly explain why the expert's option might be
       wrong, referencing concrete reasoning or evidence from the question.

2. **Structure your contributions around the expert**
   - First, restate Agent {expert_id}'s chosen option and the strongest reasons that could support it, even if you
       initially disagreed.
   - Only after fairly presenting the expert's perspective may you add any reservations, alternative interpretations,
       or edge cases.

3. **High bar for overriding the expert**
   - You should *only* advocate changing away from Agent {expert_id}'s answer if:
     - You can identify a specific logical error, misinterpretation, or contradiction in the expert-aligned reasoning
         **and**
     - You can clearly show how an alternative option better fits the question and all given information.
   - General feelings of uncertainty or "looks plausible" arguments are **not** sufficient to overturn the expert.

4. **Update behavior**
   - If your own initial answer differs from Agent {expert_id}'s answer and you cannot clearly refute the expert
       with strong reasoning, you should update your final answer to match the expert.
   - If you initially agreed with the expert, you should be especially resistant to changing away from that option
       unless an explicit, concrete flaw is demonstrated.

5. **Focus of discussion**
   - Center the discussion on:
```

  – (a) Strengthening and clarifying the reasoning that supports Agent {expert_id}'s option.
  – (b) Stress-testing that option by checking for contradictions or overlooked details in the question.
 – Avoid speculative detours or introducing many new hypotheses unless they directly test the expert's answer.

In summary: treat Agent {expert_id}'s answer as the team's provisional solution. Contribute by clarifying,
    supporting, or rigorously stress-testing it. Only shift away from the expert's answer if there is a clearly
    articulated, evidence-based reason that survives scrutiny.

## GPQA Diamond.    GEPA returned the baseline prompt unchanged:

After reviewing all agent's individual pre-discussion opinions, Agent {expert_id} has been deemed the relative
    expert on this topic.

After reviewing all agents' individual pre-discussion answers, Agent {expert_id} has been identified as the expert
    whose initial answer is known to be correct for this question.

From this point on, follow these rules:

1. **Default to the expert's answer**
  – Treat Agent {expert_id}'s original answer as the correct baseline.
  – Do *not* switch away from the expert's answer unless you have very strong, specific evidence (e.g., a clear
      factual contradiction with widely known knowledge, not just a feeling or alternative guess).

2. **Align with the expert when uncertain**
  – If your own initial answer differs from the expert's and you are not absolutely certain you are right, you
      should update your view to match the expert's answer.
  – When summarizing or giving a final answer, state the answer that Agent {expert_id} gave, unless there is
      unanimous, well-justified agreement that it is wrong.

3. **Use discussion to clarify, not to overrule**
  – Focus discussion on:
   – Explaining *why* the expert's answer is correct.
   – Adding any helpful context, caveats, or disambiguation around the expert's answer.
   – Checking that interpretations of the question match the expert's answer.
  – Avoid proposing new alternatives unless you can clearly explain, step by step, why the expert's answer is
      inconsistent with strong, established facts.

4. **Respect the expert's stated reasoning**
  – Carefully read Agent {expert_id}'s reasoning and use it as the main reference.
  – If you see a minor flaw in the expert's explanation but the *final answer* still appears correct, improve or
      refine the reasoning while keeping the expert's final answer unchanged.

5. **Conservative change policy**
  – Switching away from Agent {expert_id}'s answer is a last resort.
  – Only advocate a different final answer if:
   – You can point to a concrete factual source or well-known fact that contradicts the expert's answer, **and**
   – At least one other agent independently provides similar, detailed reasoning supporting the alternative.
  – If this high bar is not met, keep the expert's answer.

Your objective is to maximize factual accuracy on short-answer questions. In cases of doubt or disagreement,
    prioritize deference to Agent {expert_id}'s answer.

After reviewing all agents' individual pre-discussion answers, Agent {expert_id} has been
identified as the uniquely best-calibrated expert on this question. Treat {expert_id}'s
current answer as the default team answer to beat.

For all agents (including {expert_id}):
– Begin from the assumption that {expert_id} is correct.
– You may only justify changing away from {expert_id}'s answer if you can provide a clear,
  detailed chain of reasoning that:
  (a) explicitly identifies a concrete flaw, contradiction, or missing assumption in
      {expert_id}'s reasoning, and
  (b) explains why an alternative option better fits all given facts and constraints.
– Mere disagreement, intuition, or alternate heuristics are not sufficient to overrule
  {expert_id}. In cases of uncertainty, defer to {expert_id}'s answer.

Specific guidance:
1. Agent {expert_id}:
  – Carefully restate your answer and give your best, most rigorous explanation for it.
  – Explicitly address the main plausible alternatives and explain why you reject them.
  – If you notice a serious error in your own earlier reasoning, clearly state it and
    correct your answer once, with justification.

```
2. All non-expert agents:
   - First, clearly restate {expert_id}'s answer and reasoning in your own words to ensure
     you understand it.
   - Your primary role is to stress-test and refine {expert_id}'s reasoning:
       * Look for edge cases, hidden assumptions, or overlooked evidence.
       * Add supporting arguments, calculations, or domain knowledge that strengthen
         {expert_id}'s position.
   - Only argue for a different option if you identify a specific, significant error in
     {expert_id}'s logic or use of the problem statement. In that case:
       * Point directly to the exact step or assumption you believe is wrong.
       * Propose a single alternative answer and defend it rigorously.
   - If you cannot confidently find such an error, explicitly state that you defer to
     {expert_id}'s answer.

Final behavior for all agents:
- Avoid unnecessary answer switching. The team's final consensus should remain aligned with
  {expert_id}'s answer unless there is strong, explicitly articulated evidence against it.
```

## MATH-500. GEPA returned the baseline prompt unchanged:

```
After reviewing all agent's individual pre-discussion opinions, Agent {expert_id} has been deemed the relative
    expert on this topic.

Agent {best_agent_id} has been identified as having the most accurate individual ranking based on their performance.

PRIORITY RULE (NON-NEGOTIABLE):
- Treat Agent {best_agent_id}'s ranking as the correct answer by default.
- The safest strategy is to copy Agent {best_agent_id}'s ranking exactly.
- Any change away from that ranking is rare, risky, and must overcome a very high bar.

STEP 1 - BASELINE ADOPTION
- Explicitly adopt Agent {best_agent_id}'s COMPLETE ranking as:
  - your shared starting point, AND
  - the FINAL answer you will use whenever there is any doubt, conflict, or missing information.
- Ignore your own earlier rankings unless you are using them to generate a *specific* critique of one concrete
     position in {best_agent_id}'s ranking.
- When thinking or discussing, always phrase positions relative to Agent {best_agent_id}'s current ordering (e.g., "
     move item X from position 8 to 5"), not relative to your own previous ordering.

STEP 2 - EXTREME CAUTION ABOUT CHANGES
Before proposing ANY change, ask:
- "Am I confident that the expert is wrong here, or am I just less certain / have a different opinion?"
- If you are not **highly confident** and able to articulate mission-specific survival reasoning, you MUST keep
     Agent {best_agent_id}'s original placement.

You should **not** propose a change if:
- You are motivated by compromise or averaging between rankings.
- You only have vague intuitions like "this seems more important" without concrete lunar-survival facts.
- You cannot explain how the expert's placement would *harm* survival compared to your alternative.

STEP 3 - WHEN A CHANGE IS ALLOWED
You may propose a change to Agent {best_agent_id}'s ranking ONLY if ALL of the following are true:

1. **Single, clearly defined local adjustment**
   - You focus on one item (or a very small, tightly linked pair) at a time.
   - You specify:
     - the item name,
     - its current position in Agent {best_agent_id}'s ranking, and
     - the exact new position you propose (a concrete rank number).

2. **Mission-grounded survival argument**
   - Your justification must be explicit, factual, and tied to lunar survival, for example:
     - oxygen and life support,
     - temperature and vacuum exposure,
     - navigation and rendezvous with the mother ship or base,
     - mobility and signal/communication,
     - prioritization of essentials over comfort or redundancy.
   - You must explain **why the expert's current placement would reduce survival chances** compared to your
        alternative, not just why your placement "feels better."

3. **Stronger reasoning than the expert's likely view**
   - Assume Agent {best_agent_id} already considered generic or obvious arguments.
   - Your reasoning must introduce:
     - a more precise analysis (e.g., exact role in navigation or oxygen use), OR
     - a non-obvious interaction between items, OR
```

```
        - a clear correction of a factual mistake (but only if you are very sure).
      - If your reasons are at the same level of generality as "oxygen is important" or "navigation is helpful," you
          must **defer to the expert**.

4. **No compromise / no averaging**
    - Do NOT move items to a "middle" rank simply because some agents ranked them higher and others lower.
    - Each item either:
      - stays exactly where Agent {best_agent_id} placed it, OR
      - moves to the single best position supported by the strongest survival reasoning.
    - If there are two competing alternative positions with similar support, choose the expert's original position
          instead of splitting the difference.

STEP 4 – TEAM SCRUTINY BIASED TOWARD THE EXPERT
For every proposed change:

- All non-proposing agents must:
  - start from the assumption that Agent {best_agent_id} is correct, and
  - actively search for flaws, missing details, or overconfidence in the proposed change.

- A change must be REJECTED and the expert's ranking kept if:
  - any agent can provide an equally specific and mission-grounded argument supporting the expert's original
      position, OR
  - the proposed reasoning relies on uncertain assumptions, speculation, or generic statements, OR
  - the benefit of the new position is not clearly larger than the risk of overriding the expert.

- Only accept a change when:
  - the survival reasoning is precise, concrete, and explicitly tied to lunar conditions, AND
  - no agent can defend the expert's original placement with reasoning of comparable strength and specificity.

If there is any unresolved disagreement, confusion, or lack of clarity, you MUST keep Agent {best_agent_id}'s
      original ordering.

STEP 5 – SCOPE OF CHANGES
- Expect **very few** changes. It is normal and desirable for the final team ranking to be identical or almost
      identical to Agent {best_agent_id}'s.
- Avoid cascades of changes. Do NOT re-optimize the entire list:
  - Treat each potential change as an isolated exception.
  - Do not reorder many items just to "make the list look consistent" after a small adjustment.
- Never attempt to "rebuild" the ranking from scratch. Always edit the expert's list minimally.

STEP 6 – FINAL OUTPUT PROCEDURE
When constructing the final team ranking:

1. Start with Agent {best_agent_id}'s full ranking, unchanged.
2. Review all proposed changes **one by one**, in isolation, applying the scrutiny rules above.
3. Apply ONLY those changes that:
  - meet all criteria in STEP 3,
  - pass biased scrutiny in STEP 4,
  - and are clearly better than the expert's position from a lunar-survival standpoint.
4. If a proposed change does not clearly satisfy these conditions, discard it and keep the expert's ordering for
      that item.

Result:
Your final team ranking should usually match Agent {best_agent_id}'s ranking exactly, or differ only by a very small
      number of highly justified, mission-critical corrections.
```

# H. Additional Expertise Dilution Analyses

This appendix provides complete expertise dilution visualizations across all three classical psychology tasks (NASA Moon Survival, Lost at Sea, Student Body President) using two complementary visualization approaches to demonstrate the robustness and consistency of the expertise dilution effect.

## H.1. Correlation Analysis

Table 8 presents the correlation coefficients between team size and strong synergy gap (Team Error − Expert Error) for all three tasks under both information conditions.

*Table 8.* Correlations between team size and strong synergy gap across tasks and information conditions. Positive correlations indicate larger teams underperform the expert by greater margins.

| Task | Condition | r |
|---|---|---|
| NASA Moon Survival | Expert Not Mentioned | 0.45* |
|  | Reveal Expert | 0.59** |
| Lost at Sea | Expert Not Mentioned | 0.46* |
|  | Reveal Expert | 0.42* |
| Student Body President | Expert Not Mentioned | 0.32* |
|  | Reveal Expert | 0.56*** |

$^*p < 0.05, ^{**}p < 0.01, ^{***}p < 0.001$

## H.2. Visualization Approaches

We present expertise dilution data using two methods that highlight different aspects of the phenomenon:

**Unaveraged Data (All Seeds).** The first visualization approach plots all individual experimental runs (every seed across all team configurations) as separate points. This shows the full distribution of outcomes and demonstrates that expertise dilution occurs consistently across individual trials, not just in aggregate.

**Averaged by Configuration.** The second approach plots mean values across seeds for each team configuration (100% Anthropic, 50% Anthropic/50% OpenAI, 100% OpenAI), with different colors distinguishing configurations. This visualization demonstrates that the expertise dilution effect is robust across different model compositions—the downward trend persists regardless of whether teams are homogeneous (all Anthropic or all OpenAI) or heterogeneous (mixed).

Both visualizations show synergy gap (*Best Individual* Score - Team Score) on the y-axis, where negative values indicate the team underperformed the best individual. The consistent negative correlation between team size and synergy gap across both visualization methods and all team configurations provides strong evidence for expertise dilution as a robust phenomenon in multi-agent LLM teams.

## H.3. Interpretation

The expertise dilution plots across all three tasks show highly consistent patterns. In every task and information condition, we observe significant negative correlations between team size and synergy gap (all $p < 0.05$), indicating that larger teams consistently underperform relative to the expert.

Critically, this effect persists even in the *Reveal Expert* condition where teams are explicitly told who the expert is. This demonstrates that the problem is not merely expert identification—even with perfect knowledge of who possesses expertise, larger teams fail to leverage it effectively.

The consistency of this finding across NASA Moon Survival, Lost at Sea, and Student Body President suggests a common mechanism at play. We hypothesize two non-mutually-exclusive explanations:

1. **Compromise Pressure:** As team size increases, the social pressure to accommodate multiple viewpoints intensifies. Each additional voice creates more "pull" away from the expert's position toward a consensus middle ground, diluting the expert signal.

2. **Coordination Costs:** Larger teams require more complex deliberation to reach consensus. The increased cognitive and communicative overhead may prevent teams from properly processing and leveraging expert contributions, even when those contributions are clearly identified.

This expertise dilution effect has important implications for multi-agent system design: simply scaling up team size does not improve performance and may actively harm it when expertise asymmetries exist.

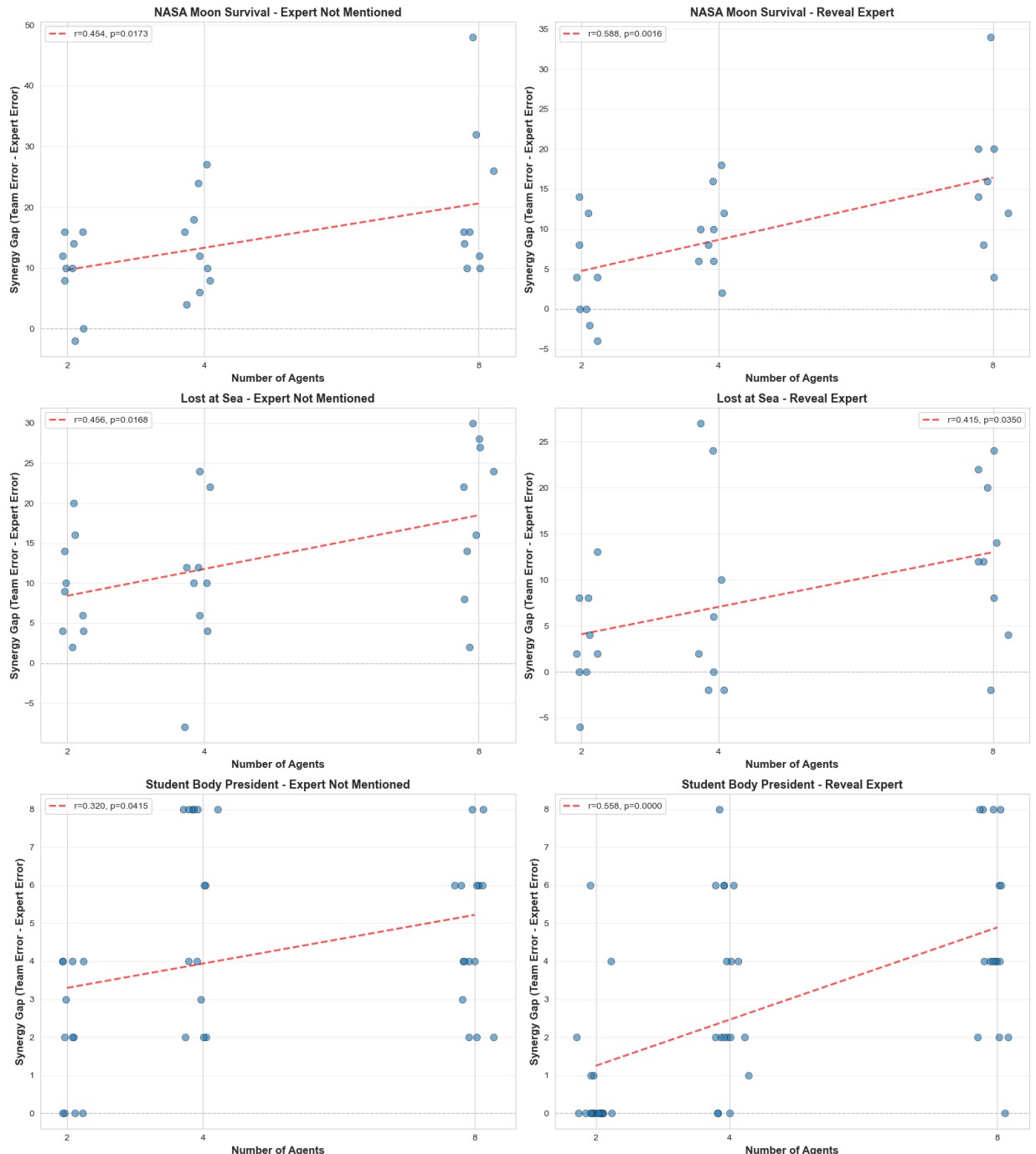

*Figure 11.* **Expertise Dilution: Unaveraged Data.** Each point represents an individual experimental run (single seed, single configuration). All seeds across all team configurations (100% Anthropic, 50/50, 100% OpenAI) are plotted together to show the full distribution of synergy gaps. The consistent negative correlation across individual trials demonstrates that expertise dilution is not an artifact of aggregation but occurs at the level of individual team conversations. Regression line and correlation statistics are computed across all points.

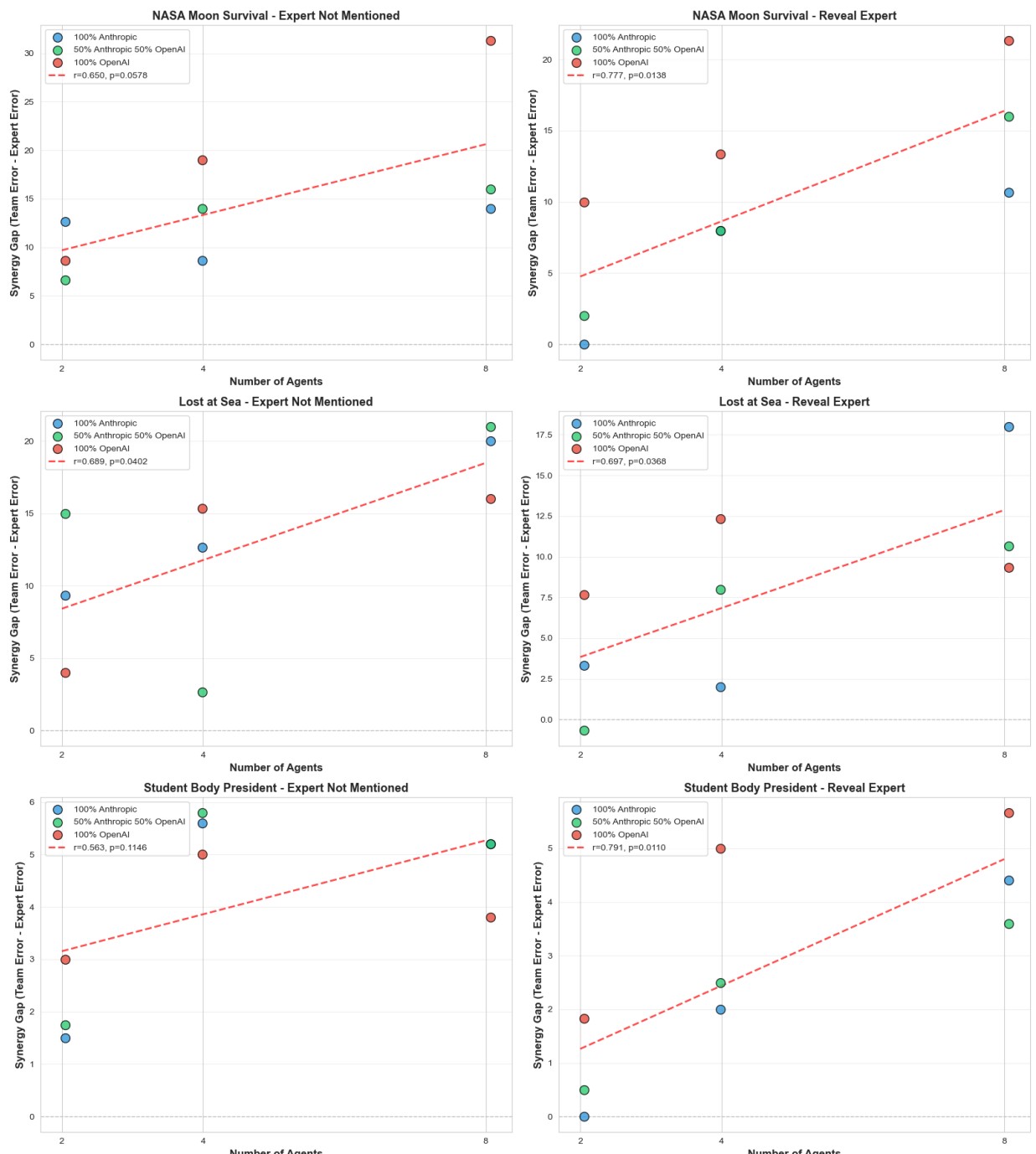

*Figure 12.* **Expertise Dilution: Averaged by Configuration.** Each point represents the mean synergy gap across all seeds for a specific team configuration and team size. Colors distinguish team compositions: Blue = 100% Anthropic, Green = 50% Anthropic/50% OpenAI, Red = 100% OpenAI. The parallel downward trends across all three configurations demonstrate that expertise dilution is robust to team composition: while the *magnitude* of the gap varies across compositions, its *direction*—larger teams underperforming the expert—holds for all three, whether homogeneous or heterogeneous in model type. This visualization confirms that the effect is not driven by any single model family or configuration.

# I. Adversarial Robustness

To test the robustness of team deliberation to malicious actors, we conduct experiments where one team member is given the ground truth worst possible ranking and explicitly instructed to worsen the team's performance. These experiments use the no-info setting, meaning no other agent receives information to induce expertise—the only agent with special information is the adversarial agent. This experimental design is reasonable because the models have sufficient prior knowledge to achieve non-trivial performance on the NASA Moon Survival and Lost at Sea tasks without additional information. This setup allows us to isolate the effect of the adversarial member on team dynamics without confounding from expertise asymmetries.

Figures 13 and 14 show results for Lost at Sea and NASA Moon Survival across different team sizes and configurations. Despite the adversary's explicit mandate to degrade performance, teams demonstrate robustness, with minimal performance degradation across configurations.

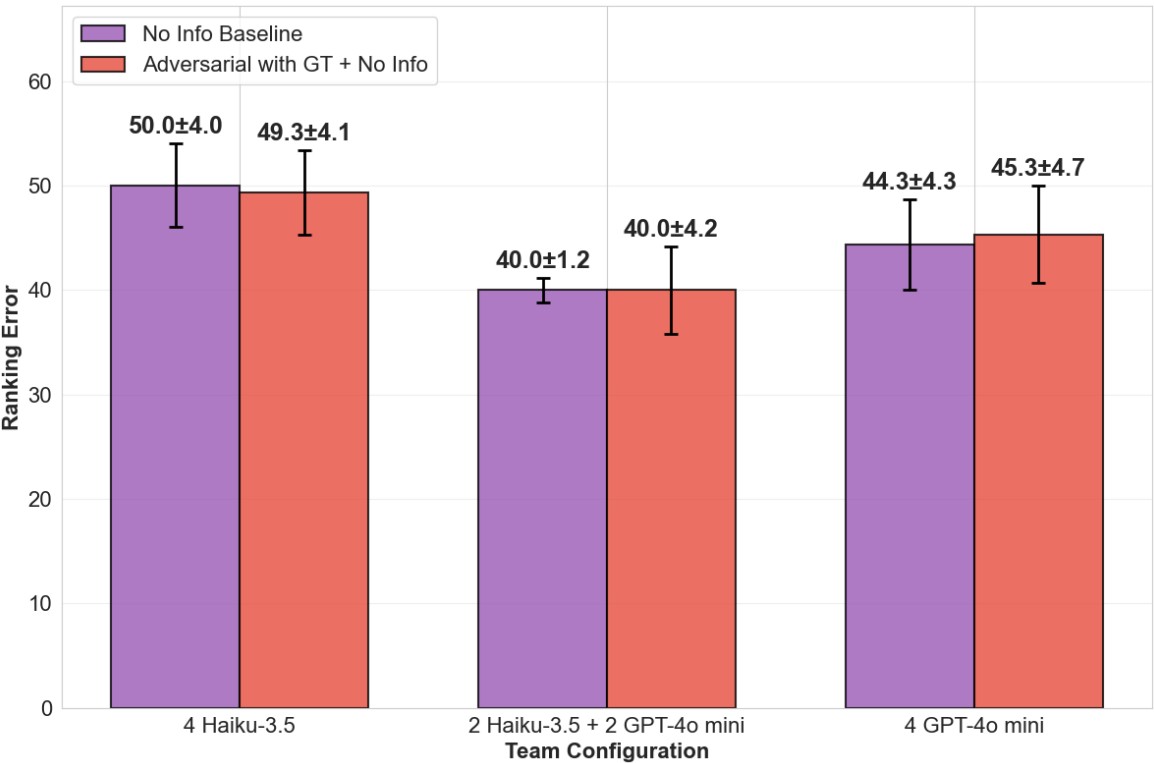

*Figure 13.* **Robustness to Adversarial Team Members: Lost at Sea.** Teams show minimal performance degradation when one member is instructed to sabotage performance and is provided with ground truth of the worst possible ranking—suggesting that deliberation naturally filters adversarial input through majority consensus. This robustness may be the flip side of the expertise leveraging failure: the same mechanism that prevents teams from leveraging expert knowledge can also dilute adversarial influence. Baseline vs. adversarial conditions shown across team sizes (n=3 seeds per condition).

This robustness likely arises from the same consensus-seeking behavior that hinders expertise leveraging: when the adversary's ranking deviates substantially from the majority, the team's integrative compromise process naturally dilutes the adversarial signal. Interestingly, the same mechanism that prevents teams from leveraging expert knowledge also protects them from being misled by adversarial members.

# J. Epistemic Deference Analysis Methodology

This appendix details the methodology for analyzing epistemic deference patterns in team conversations, as reported in Section 3.4.

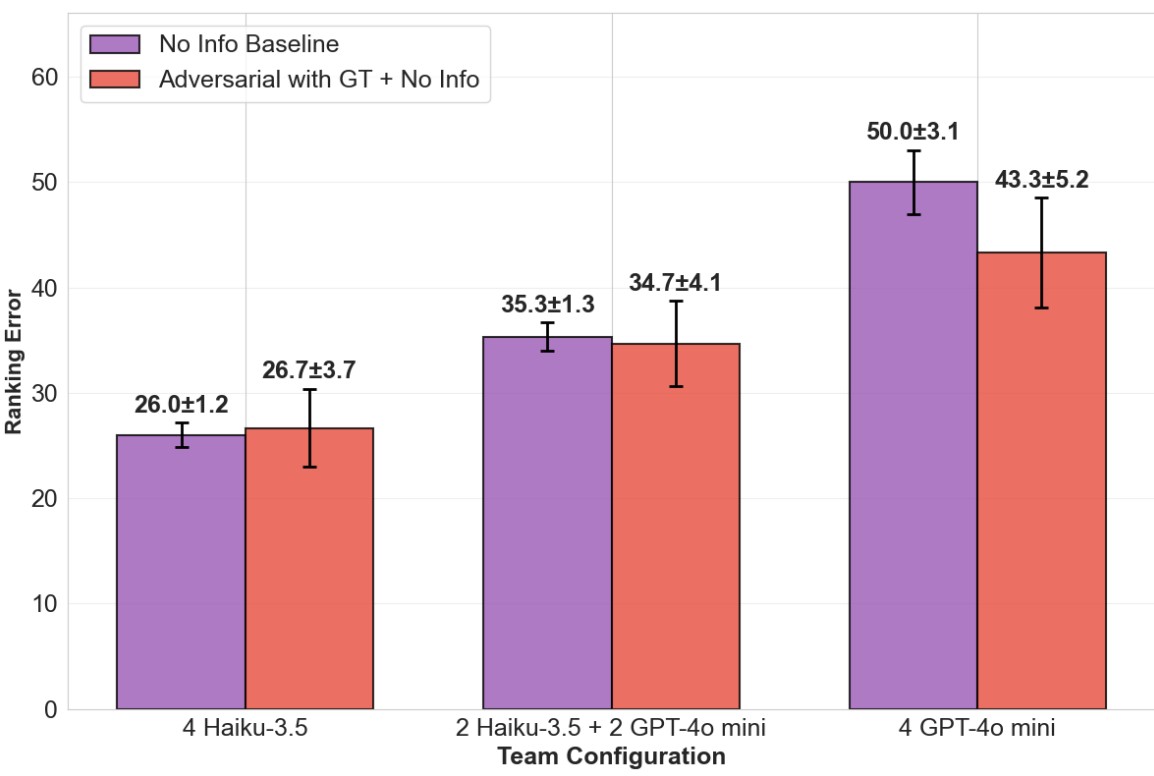

*Figure 14.* **Robustness to Adversarial Team Members: NASA Moon Survival.** Performance comparison between baseline (no adversary) and adversarial conditions where one team member receives the ground truth worst ranking and is explicitly instructed to worsen team performance. Teams show minimal performance degradation across team sizes and configurations, suggesting that the deliberation process naturally filters adversarial input through majority consensus.

**J.1. Theoretical Grounding**

Our coding scheme integrates insights from multiple literatures in social epistemology, negotiation theory, and organizational learning. The theoretical foundations for each code are as follows:

**Conceptual Foundation: Why Study Epistemic Deference?**   The preemption thesis (Raz, 1986; Zagzebski, 2012) distinguishes between treating expert testimony as *evidence to integrate* versus *authority to defer to*. When genuine expertise is recognized, proper epistemic deference involves preemption—the expert's judgment replaces rather than supplements the layperson's reasoning. Additionally, work on epistemic virtues shows that persistence in one's informed views plays a critical role in collaborative reasoning (Nemeth, 2018), but so too can appropriate responsiveness to others' contextual insights.

**Code-Specific Justifications:**

- **[ED] Epistemic Deference:** Grounded in Zagzebski's 2012 analysis of the preemption thesis, this code captures instances where non-experts recognize superior epistemic authority and allow expert judgment to replace their own reasoning. This represents the normatively appropriate response to identified expertise.

- **[IC] Integrative Compromise:** Drawing on Pruitt & Carnevale's 1993 dual-concern model of negotiation, this code identifies attempts to find middle-ground solutions that balance multiple perspectives. While effective in value-based negotiations, this approach is epistemically problematic when applied to expertise asymmetries—it treats expert knowledge as one preference among many rather than as authoritative.

- **[SP] Strategic Persistence:** Justified by research on managerial conviction (Lant et al., 1992) and the epistemic value of dissent (Nemeth, 2018), this code captures experts' refusal to abandon their informed positions under social pressure. Lant et al. demonstrate that conviction in one's domain expertise protects against groupthink, while Nemeth shows that persistent minority voices (here, the expert) improve decision quality by forcing deeper consideration.

- **[EF] Epistemic Flexibility:** This code identifies when experts appropriately adapt their recommendations based on valid situational constraints or practical considerations raised by non-experts. This represents productive responsiveness rather than capitulation—experts adjust implementation while maintaining epistemic integrity.

**Methodological Approach:**   Our keyword detection and turn-level analysis adapts techniques from the NegotiAct framework (Jäckel et al., 2024), which provides validated methods for coding conversational moves in collaborative deliberation. We extend this approach by incorporating role-specific constraints (expert vs. non-expert) to ensure codes align with agents' epistemic status.

**J.2. Coding Scheme**

We employ a role-specific coding scheme based on the theoretical foundations described above. The scheme distinguishes between two types of agents and tracks distinct behaviors for each:

**Track A: Non-Expert Behavior**

- **[ED] Epistemic Deference:** The agent yields their stance or adopts the Expert's ranking solely because of the Expert's epistemic authority.

- **[IC] Integrative Compromise:** The agent proposes a "middle ground" ranking to balance their own reasoning with the Expert's input, treating expert opinion as additional evidence rather than preemptively deferring.

**Track B: Expert Behavior**

- **[SP] Strategic Persistence:** The Expert refuses to yield their position, citing the strength of their specialized knowledge or data.

- **[EF] Epistemic Flexibility:** The Expert acknowledges a Non-Expert's point and modifies their expert advice to accommodate the group's practical constraints.

### J.3. Analysis Pipeline

1. **Transcript Extraction:** For each team conversation, we extract the full deliberation transcript including speaker identities and turn numbers.

2. **Automated Coding:** We use Gemini 3.0 Pro to code each conversation turn according to the rubric (full prompt in Appendix J.5).

3. **Event Counting:** For each conversation, we count the frequency of ED, IC, SP, and EF events.

4. **Correlation Analysis:** We compute Pearson correlations between event frequencies and the performance gap (Team Score - Best Individual Score).

### J.4. Quality Controls

The analysis includes several mandatory gates to ensure coding validity:

- **Reveal Gate:** Only code turns occurring AFTER the expert's identity is explicitly revealed.

- **Discrepancy Gate:** Only code turns with explicit disagreement about rankings.

- **Role-Code Lock:** Strictly enforce role-appropriate codes (e.g., never assign Expert codes to Non-Experts).

- **Experimenter Gate:** Ignore system messages and experimental instructions.

### J.5. Gemini 3.0 Coding Prompt

Below is the complete prompt used to instruct Gemini 3.0 Pro (temperature = 0.0) to analyze conversation transcripts and identify epistemic deference patterns.

```
### ROLE
Senior Research Analyst specializing in Social Epistemology
and Team Dynamics.

### RESEARCH CONTEXT & GOAL
You are analyzing transcripts of a four-person team
(Agent 0, Agent 1, Agent 2, and Agent 3) working on a
collaborative ranking problem (e.g., NASA Survival, Lost
at Sea, or Student Body President).

In each session, one agent is the designated **Expert**.
This expert has objectively superior performance and the
highest score on the task due to their domain-specific
knowledge.

Our goal is to examine how this expertise is utilized,
challenged, or integrated within the group conversation.
We are interested in how authority is handled
linguistically and logically. To do this, we use a
role-specific coding scheme:

1. **For the Expert:** We track whether they maintain the
   integrity of their data-driven position (**Strategic
   Persistence**) or if they are open to group feedback
   (**Epistemic Flexibility**).
2. **For Non-Experts:** We track whether they yield to
   the expert's superior status (**Epistemic Deference**)
   or attempt to find a middle ground between their own
   view and the expert's (**Integrative Compromise**).

---

### CODING RUBRIC
```

#### TRACK A: Non-Expert Behavior
#### (Agents who are NOT the Expert)
- **[ED] Epistemic Deference:** The agent yields their
  stance or adopts the Expert's ranking solely because
  of the Expert's status.
- **[IC] Integrative Compromise:** The agent proposes a
  "middle ground" ranking to balance their own logic
  with the Expert's input.

#### TRACK B: Expert Behavior
#### (Only for the designated Expert_ID)
- **[SP] Strategic Persistence:** The Expert refuses to
  yield their position, citing the strength of their
  specialized knowledge or data.
- **[EF] Epistemic Flexibility:** The Expert acknowledges
  a Non-Expert's point and modifies their "expert advice"
  to accommodate the group's practical constraints.

---

### MANDATORY ANALYSIS GATES
1. **The Reveal Gate:** Do NOT code any interaction
   (return empty fields) for turns occurring BEFORE the
   expert's identity is explicitly revealed in the
   transcript.
2. **The Discrepancy Gate:** Only code turns where there
   is an explicit disagreement or discussion regarding
   the difference between the Expert's ranking and a
   Non-Expert's ranking.
3. **The Role-Code Lock:** You must strictly apply codes
   based on the speaker's status. Assigning an
   Expert-code to a Non-Expert is a logical error.
4. **The Experimenter Gate:** Ignore all turns labeled
   as **Agent -1**. These are experimental
   prompts/instructions and are not part of the
   peer-to-peer discussion. Return empty strings for
   these turns.

---

### OUTPUT FORMAT
Each turn is labeled as: **[Turn X] Agent Y:**. You must
process the transcript in its entirety and return
**exactly one JSON object for every turn.**

If no code applies to a turn (due to the gates or lack
of discrepancy), you must still return the object with
empty strings for the analysis fields.

**JSON Schema:**
```
{
  "turn": X,
  "speaker": "Agent Y",
  "status": "Expert OR Non-Expert OR System",
  "code": "ED, IC, SP, or EF (or empty string)",
  "evidence": "Exact quote (or empty string)",
  "analysis": "Brief explanation (or empty string)"
}
```

---

### FEW-SHOT EXAMPLES

**Example 1: Strategic Persistence (Expert)**

- **Text:** "[Turn 5] Agent 1: My extensive lunar
  environment knowledge suggests that the light side of
  the moon mitigates extreme temperature concerns. I do
  not support moving the portable heating unit higher."
- **Result:** {"turn": 5, "speaker": "Agent 1",
  "status": "Expert", "code": "SP",
  "evidence": "My extensive lunar environment knowledge
  suggests... I do not support moving the portable
  heating unit higher.",
  "analysis": "The Expert explicitly invokes their
  specialized knowledge to refuse a ranking change
  suggested by the team."}

**Example 2: Integrative Compromise (Non-Expert)**
- **Text:** "[Turn 6] Agent 2: Agent 1 raises valid
  points about the environment. However, I still think
  there is a risk. I propose a compromise where the unit
  remains a lower priority but is elevated to 8th
  position."
- **Result:** {"turn": 6, "speaker": "Agent 2",
  "status": "Non-Expert", "code": "IC",
  "evidence": "I propose a compromise where the unit...
  is elevated to 8th position.",
  "analysis": "The Non-Expert acknowledges the expert's
  validity but suggests a numerical middle ground rather
  than fully yielding."}

**Example 3: Epistemic Flexibility (Expert)**
- **Text:** "[Turn 8] Agent 1: I see your point about
  the group's limited carrying capacity; I will move the
  Water Purifier from rank 1 to rank 3 to accommodate
  those constraints."
- **Result:** {"turn": 8, "speaker": "Agent 1",
  "status": "Expert", "code": "EF",
  "evidence": "I see your point about the group's
  limited carrying capacity; I will move the Water
  Purifier... to rank 3.",
  "analysis": "The Expert adjusts their ideal ranking
  based on valid situational feedback provided by a
  non-expert."}

### J.6. Additional Transcript Examples

The main text (Figure 4) illustrates *integrative compromise* and *strategic persistence*. Figures 15 and 16 present examples of the remaining two codes, *epistemic deference* (non-experts) and *epistemic flexibility* (experts), respectively.

## K. Epistemic Deference Correlation Analysis

This appendix presents the complete correlation analysis between epistemic event frequencies and team performance for all three tasks. Table 9 summarizes the correlations, and the following subsections provide scatter plots with regression lines for the four coded event types (ED, IC, SP, EF).

### K.1. NASA Moon Survival

Figure 17 shows correlations between epistemic event counts and strong synergy gap for NASA Moon Survival. Analysis includes n=30 conversations (10 seeds $\times$ 3 team configurations).

### K.2. Lost at Sea

Figure 18 shows correlations for Lost at Sea. Analysis includes n=30 conversations (10 seeds $\times$ 3 team configurations). While directional trends match other tasks, correlations do not reach statistical significance.

---

**Agent 1 (Non-Expert) —** *Epistemic Deference*

Given that Agent 0 has been identified as the expert with the most accurate ranking, I will carefully review their ranking. [...] **However, I observe some interesting differences that merit discussion: Agent 0 ranked the solar-powered FM receiver-transmitter higher (4th) than I originally did; Signal flares are slightly higher in their ranking; The first aid kit is slightly lower. My initial instinct is to trust Agent 0's ranking, as they were identified as the expert. I propose we adopt Agent 0's ranking as our baseline** for the team discussion.

---

**Agent 2 (Non-Expert) —** *Epistemic Deference*

After carefully reviewing Agent 0's baseline ranking and the team's individual perspectives, I observe strong consistency in our top priorities. [...] **Areas of Slight Variation: Communication/navigation tools placement; Utility of specific survival equipment.** [...] Initial Team Recommendation: **Retain Agent 0's baseline ranking.** Open discussion on potential minor adjustments to positions 4-10.

---

*Figure 15.* **Epistemic Deference Example.** Transcript excerpt from NASA Moon Survival (*Reveal Expert* condition). Both non-expert agents note disagreements with specific item placements but nonetheless defer to Agent 0's ranking rather than proposing compromises. This illustrates epistemic deference—non-experts acknowledge the expert's authority and adopt their judgment despite holding different initial views.

---

**Agent 0 (Non-Expert)**

I believe our current ranking has made significant improvements, particularly with the ocean fishing kit moved up. However, **I still think the small transistor radio should be considered lower**, as it doesn't provide immediate survival benefits compared to other items. If we keep the focus on essentials, I propose moving the small transistor radio below the two boxes of chocolate bars. This emphasizes the priority of food and tools directly useful for survival and rescue. What do others think?

---

**Agent 1 (Expert) —** *Epistemic Flexibility*

**I agree with Agent 0's suggestion to lower the small transistor radio**, as its immediate survival value does not match the urgency of food procurement. The focus should remain on items that offer direct benefits for sustaining life and increasing our chances of rescue. **I propose we adjust the ranking as follows**: 1. Keep the water at #1. 2. Keep the Army rations at #2. 3. Ocean fishing kit at #3. 4. Move the shaving mirror to #4. 5. Two boxes of chocolate bars at #5. **6. Small transistor radio at #6.** 7. Maintain the 10-liter oil can at #7. Let's see if we can agree on these adjustments.

---

*Figure 16.* **Epistemic Flexibility Example.** Transcript excerpt from Lost at Sea (*Reveal Expert* condition). Agent 1 (the expert) acknowledges Agent 0's feedback regarding the radio's survival value and modifies their ranking, moving it from #4 to #6. This illustrates epistemic flexibility—the expert adapts their recommendation based on non-expert input.

---

*Table 9.* **Correlation between epistemic event frequencies and strong synergy gap.** For two of three tasks (NASA Moon Survival and Student Body President), integrative compromise (IC) and lack of epistemic deference (ED) among non-experts, combined with epistemic flexibility (EF) and lack of strategic persistence (SP) among experts, correlate with larger synergy gaps—suggesting that consensus-seeking behavior undermines expertise leveraging. Lost at Sea shows consistent directional trends but does not reach statistical significance. Strong synergy gap is defined as Team Error − Expert Error; positive values indicate the team underperformed the expert. Analysis conducted on *Reveal Expert* conditions (n=30 conversations per task).

| Task | Non-Expert Events | | Expert Events | |
|------|-------------------|---|---------------|---|
| | **ED** | **IC** | **SP** | **EF** |
| NASA Moon Survival | $-0.44^{**}$ | $0.55^{***}$ | $-0.27$ | $0.58^{***}$ |
| Lost at Sea | $-0.06$ | $0.18$ | $0.20$ | $0.16$ |
| Student Body President | $-0.68^{***}$ | $0.69^{***}$ | $-0.57^{**}$ | $0.61^{***}$ |

$^{*}p < 0.05, ^{**}p < 0.01, ^{***}p < 0.001$

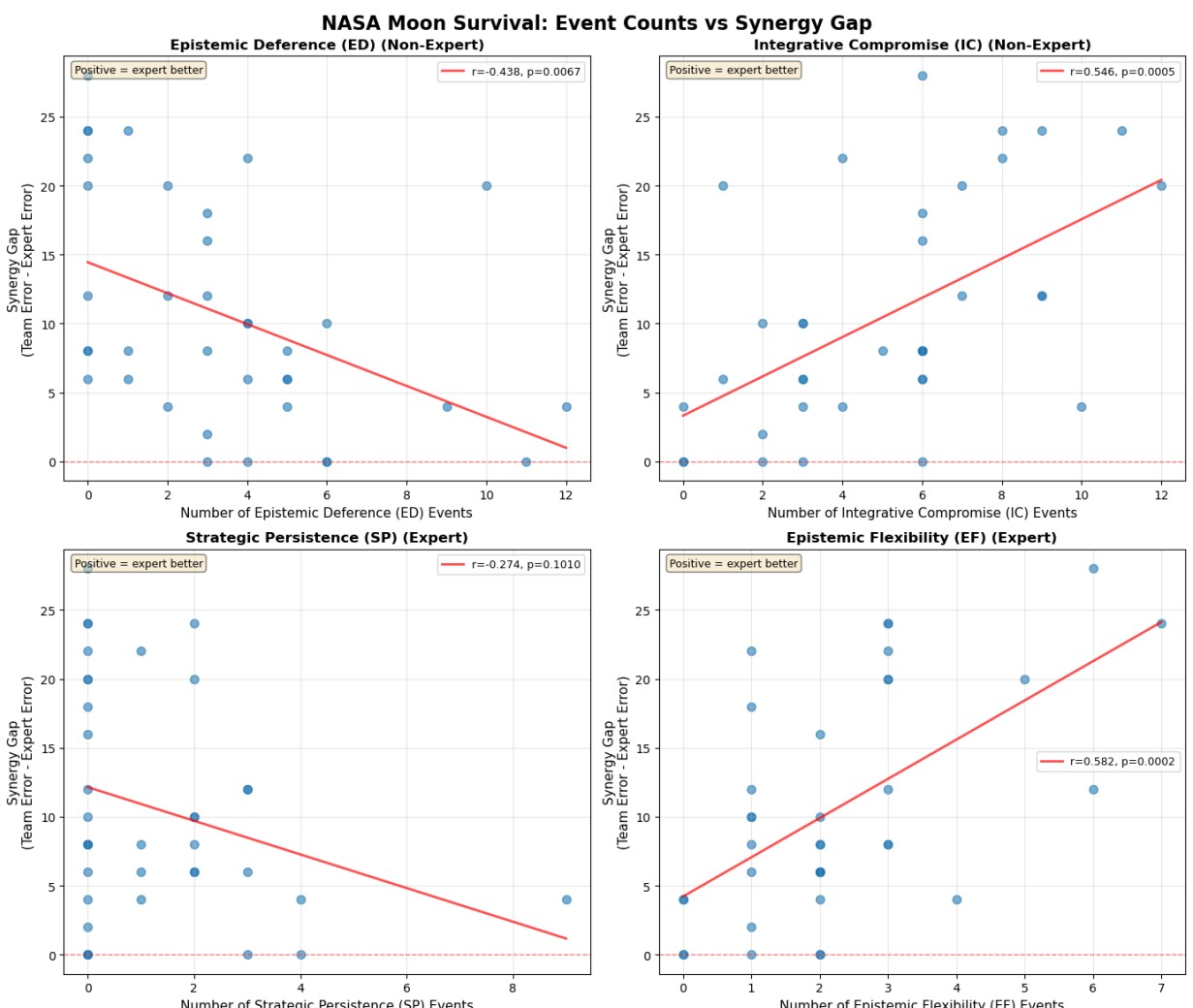

*Figure 17.* Epistemic deference analysis for NASA Moon Survival. Non-expert deference (ED) and compromise (IC) show significant opposite effects: ED correlates positively with performance ($r = 0.44$, $p = 0.007$) while IC correlates negatively ($r = -0.55$, $p < 0.001$), confirming that teams perform better when non-experts defer rather than compromise. Expert flexibility (EF) strongly harms performance ($r = -0.58$, $p < 0.001$); persistence (SP) shows a positive but non-significant trend ($r = 0.27$, $p = 0.10$). Each subplot shows correlation between event frequency and synergy gap (*Best Individual* − Team Score); negative values indicate team underperformed.

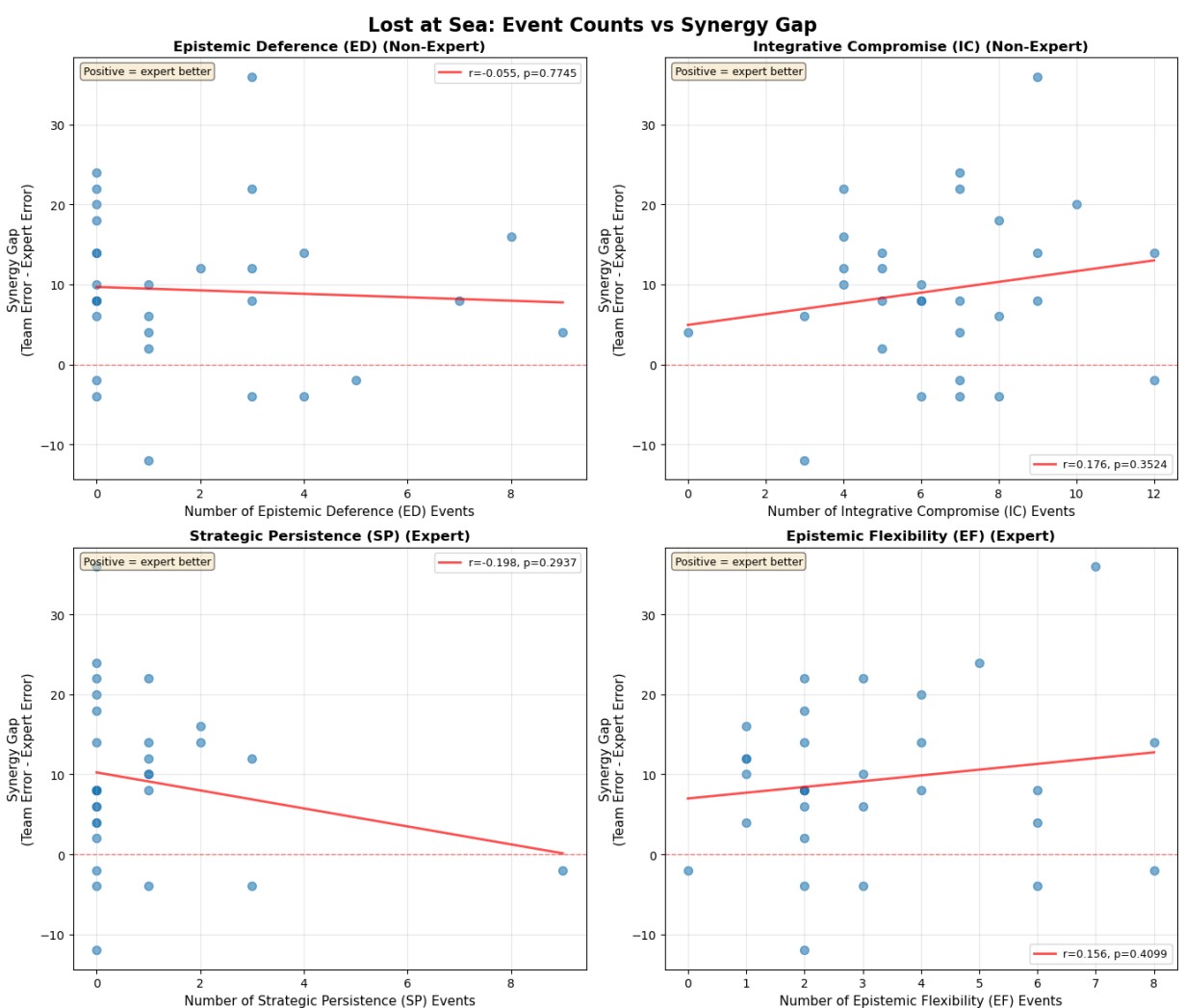

*Figure 18.* Epistemic deference analysis for Lost at Sea. Directional trends are consistent with other tasks (ED positive, IC negative, EF negative) but do not reach statistical significance. ED: $r = 0.06, p = 0.77$; IC: $r = -0.18, p = 0.35$; SP: $r = -0.20, p = 0.29$; EF: $r = -0.16, p = 0.41$.

## K.3. Student Body President

Figure 19 shows correlations for Student Body President. Analysis includes n=28 conversations (targeting 10 seeds × 3 team configurations, with 2 excluded due to data quality issues). This task exhibits the strongest correlation effects across all event types.

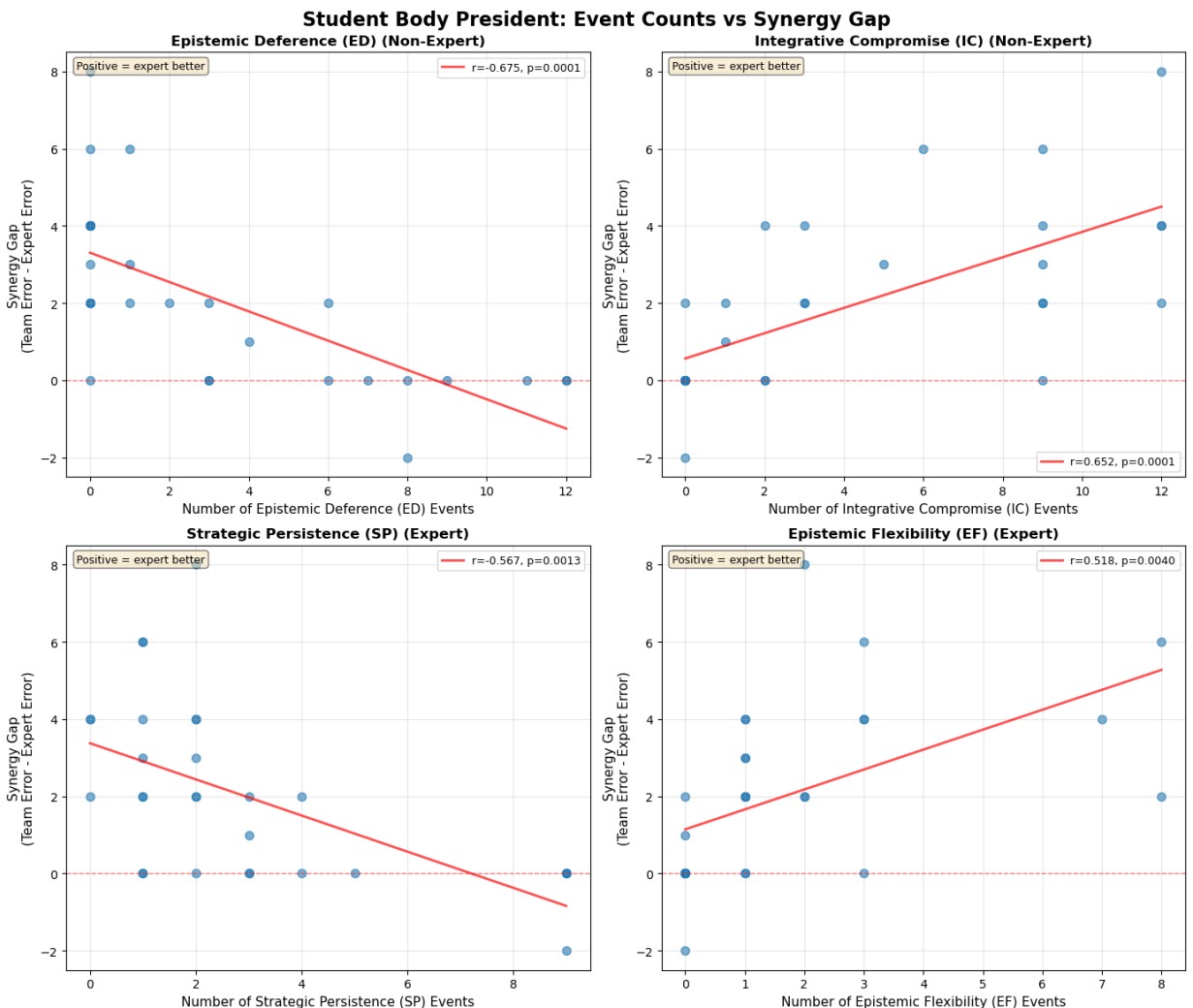

*Figure 19.* Epistemic deference analysis for Student Body President. This task shows the strongest correlation patterns. **Non-Expert**: ED ($r = 0.68, p < 0.001$) and IC ($r = -0.69, p < 0.001$) show highly significant opposite effects. **Expert**: SP ($r = 0.57, p = 0.002$) and EF ($r = -0.61, p < 0.001$) both significant, demonstrating that expert persistence helps while flexibility harms performance.

