# OpenReview forum: "Multi-Agent Teams Hold Experts Back"
_ICML.cc/2026/Conference — ICML 2026 regular_

### Official Review · Reviewer_uZ9q · 2026-03-10

**Soundness:** 3
**Presentation:** 2
**Significance:** 3
**Originality:** 3
**Overall Recommendation:** 3
**Confidence:** 3

**Summary:**

This paper studies whether self-organizing multi-agent LLM teams can achieve strong synergy, i.e., match or exceed the best individual member in the team. Across both classic organizational decision tasks and modern ML benchmarks, the authors find that teams often fail to reach expert-level performance even when the expert is explicitly revealed. The paper further argues that the main bottleneck is not expert identification but expert leveraging: teams tend to compromise across opinions rather than defer effectively to the strongest member. The paper also discusses team-size effects and the connection between this behavior and adversarial robustness.

**Compliance With Llm Reviewing Policy:**

Affirmed.

**Final Justification:**

The author's reply has already resolved my problem. I hope to keep my rating.

**Key Questions For Authors:**

1.How much of the reported gap is due to the discussion process itself versus the final-answer selection protocol? In particular, what happens if the final answer is taken directly from the revealed expert, selected by a separate judge model, or chosen by majority vote over post-discussion answers?
2.How robust are the main findings to changes in the discussion protocol, such as turn-taking constraints, or allowing only a single revision after the expert speaks?
3.In the Reveal Expert setting, can the authors include the explicit baseline where the revealed expert answers alone, without discussion?

**Limitations:**

Yes.

**Strengths And Weaknesses:**

Strengths:
1.This article carefully analyzes the behavioral mechanisms behind the results.
2.The evaluation spans both controlled psychology-style tasks and realistic frontier-model benchmarks.
3.The paper asks an important question for multi-agent LLM systems: whether teams can actually exploit heterogeneous expertise rather than merely average opinions.
Weaknesses:
1.The results are convincing for the specific free-form deliberation protocol used here, but it is less clear how far they generalize to other coordination schemes.
2.The writing and figure presentation could be improved.
3.The paper would be stronger with more competitive baselines, such as confidence-weighted aggregation, judge-based answer selection, or authority-aware coordination protocols.

---

> ### Author Rebuttal · Authors · 2026-03-31
>
> We thank the reviewer for their assessment and recognition of our "careful behavioral analysis." The reviewer's constructive suggestions — particularly on additional baselines — have meaningfully strengthened the paper, and we are grateful for the engagement. We address each question below.
>
> **Terminology.** ALOC (At Least One Correct) is the per-problem upper bound — the team gets credit if any member answered correctly; RE (Reveal Expert) = team told who the expert is; ENM (Expert Not Mentioned) = no expertise info shared (Section 3.3).
>
> **New baselines (conducted during rebuttal on HLE Text-Only):** Debate = Du et al. (2024) parallel broadcast-and-revise protocol, 4 agents, 4 rounds; Opt-Out = agents may abstain from discussion rounds; CoT+MV = independent chain-of-thought reasoning, majority vote, no interaction.
>
> | Method | Accuracy |
> |--------|:--------:|
> | ALOC (upper bound) | 45% |
> | Team (RE) | 34% |
> | Opt-Out (RE) | 31% |
> | Opt-Out (ENM) | 31% |
> | Team (ENM) | 28% |
> | Debate (ENM) | 26% |
> | Debate (RE) | 23% |
> | CoT + Majority Vote | 14% |
>
> All methods fall 11-31pp short of ALOC.
>
> **Q1: "How much of the gap is due to the discussion process itself versus the final-answer selection protocol? What if the final answer is taken from the revealed expert, by majority vote, or by a judge?"**
>
> Great questions. We now report team accuracy as the majority vote of all agents' post-discussion individual answers. Across all benchmarks (old→MV, ENM/RE): MMLU Pro 85→86/89→89, GPQA 74→76/82→83, SimpleQA 50→51/54→54, HLE 29→28/35→34, MATH-500 67→63/73→71. Max shift: 4pp; agreement >97%.
>
> Taking the answer from the revealed expert directly would yield ALOC (45%) by construction — the expert is defined as whichever agent answered correctly (Section 3.3). The 11pp gap between ALOC and Team RE (34%) is precisely the expertise leveraging failure: the team has the correct answer among its members but fails to adopt it. CoT + majority vote with no discussion achieves only 14%, so discussion helps (14% → 28–34%), but cannot close the gap to ALOC. These results support the conclusion that the bottleneck is expert leveraging during deliberation, not answer selection.
>
> **Q2: "How robust are the main findings to changes in the discussion protocol, such as turn-taking constraints, or allowing only a single revision after the expert speaks?"**
>
> Thank you for raising this -- with the new experiments, we now test three fundamentally different protocols on HLE: (1) our deliberation protocol (28–34%), (2) Du et al. (2024) broadcast-and-revise debate (23–26%), and (3) opt-out allowing agents to abstain (31%). With CoT+MV (14%), all four coordination schemes fall short of ALOC. These span parallel broadcast, sequential with abstention, sequential with randomized order, and no interaction, which we believe are representative points in the multi-agent protocol design space. The paper also demonstrates robustness across 3 compositions, 2 expertise distributions, 4 information conditions, 3 team sizes, and 8 tasks.
>
> **Q3: "In the Reveal Expert setting, can the authors include the explicit baseline where the revealed expert answers alone, without discussion?"**
>
> Thank you for raising this — this baseline is already in the paper. The "Expert Baseline" / ALOC (orange bar in Figures 2, 5, 6, 7) represents exactly this: the per-problem best member's answer, no team, no discussion. Section 3.3 defines it explicitly. The team consistently underperforms this baseline, which is the core finding. We will make this easier to locate in the revision.
>
> **"The writing and figure presentation could be improved."**
>
> Thank you — we will correct the Figure 5/6 caption issues noted by another reviewer in the revision. Could the reviewer point us to specific figures or aspects of presentation they would like to see improved? We want to make sure the revision addresses your concerns directly.
>
> **"The paper would be stronger with more competitive baselines, such as confidence-weighted aggregation, judge-based answer selection, or authority-aware coordination protocols."**
>
> Thank you -- we have now added three baselines (see summary table): (1) **CoT + majority vote** (14% on HLE), (2) **Du et al. (2024) debate** (23–26%), and (3) **opt-out** (31%). Regarding confidence-weighted aggregation and judge-based selection: post-discussion agreement exceeds 97%, so any reasonable aggregation produces nearly identical results. The authority-aware protocol is essentially our RE condition (Appendix A.3). Thank you for pushing us on baselines — the paper is stronger for it.
>
> We hope these additions — particularly the three new baselines motivated by the reviewer's suggestions — address the concerns that led to a score of 3. We would be grateful if the reviewer would consider updating their assessment.
>
> We again thank the reviewer for engaging with the work and helping us strengthen the paper!

---

> > ### Author Rebuttal · Reviewer_uZ9q · 2026-04-03
> >
> > My question has been answered, and I have no further questions.

---

> > > ### Author Response · Authors · 2026-04-03
> > >
> > > Thank you for your continued engagement with the work as your feedback has helped improve the paper!
> > >
> > > We would appreciate if the reviewer would consider raising their score in light of the new results we've provided to address their questions. We appreciate their consideration.

---

### Official Review · Reviewer_6PDV · 2026-03-12

**Soundness:** 2
**Presentation:** 3
**Significance:** 3
**Originality:** 2
**Overall Recommendation:** 3
**Confidence:** 4

**Summary:**

This paper evaluates the collaborative performance of LLM agents within multi-agent teams. The setups are inspired by experimental designs used to study human-human teamwork, specifically employing expert manipulation to create information asymmetry. Unlike most research demonstrating that multi-agent setups improve an LLM agent's reasoning capabilities, this work finds that a team of LLM agents fails to outperform individual agents across both psychological team tasks and frontier reasoning benchmarks. The authors provide a broad discussion of the observed patterns, including expert dilution, resilience to adversaries, and scaling effects regarding team size and discussion rounds.

**Compliance With Llm Reviewing Policy:**

Affirmed.

**Final Justification:**

Most of my technical concerns have been addressed, I raise my score from 2 to 3.

**Key Questions For Authors:**

1. Is the information asymmetry manipulation genuinely effective?

2. Why evaluating team performance via querying individual team members?

3. Why were state-of-the-art (SOTA) models not tested on the psychological tasks?

4. How exactly are the communication logs and discussion histories stored and represented within the context window of each individual agent?

**Limitations:**

yes

**Strengths And Weaknesses:**

**Strengths**
1. This paper tackles a pressing yet underexplored research question: will LLM agents trained for one-on-one interactions perform effectively in multi-agent teams? The observed patterns of non-experts' over-compliance and experts' lack of resistance effectively capture the misaligned training goals between standard conversational chatbots and effective team players. This provides important insights for future training paradigms in multi-agent settings.

2. The research is well-rooted in theories from industrial and organizational psychology. Furthermore, the conversation analysis provides a solid qualitative explanation for the observed performance gaps.

3. The paper is well written and easy to follow. The authors perform rigorous statistical analysis to back up their claimed conclusions. Significance values and error bars are clearly presented throughout the paper.

**Weaknesses**

1. My major concern revolves around the counter-intuitive finding that multiple LLM agents in a team perform worse than individual agents. Given the enormous amount of existing work demonstrating the superiority of multi-agent LLM systems and the success of Mixture-of-Experts (MoE) architectures, I am inclined to attribute these divergent observations to the specific evaluation methods and experimental designs employed in this study. Consequently, the applicability of these findings to more generalized multi-agent LLM systems remains questionable.

2. Is the information asymmetry manipulation genuinely effective? Considering the popularity of the psychological tasks utilized (e.g., the NASA lunar survival task), it is highly probable that the ground truth answers are already exposed within the training corpora of modern LLMs. Under this assumption, manipulating information asymmetry by providing privileged information and revealing expert identities may not function as intended. Running a formal manipulation check on SOTA models to validate this setup would significantly strengthen the paper.

3. While the authors claim ecological validity for their multi-agent teamwork setups, a fundamental difference between human teamwork and the proposed environment lies in autonomy and emergent collaboration strategies. Although described as self-organizing teams, the LLM agents are constrained by a fixed speaking sequence (albeit randomized) and a fixed discussion length. Assuming each agent maintains a context memory of previous discussions and all communications are globally visible, these design decisions implicitly create a less-efficient team strategy. Specifically, it forces contributions even when agents lack expertise or when the team has already reached a consensus. It would be valuable to investigate whether this performance degradation persists in more dynamic, agentic teamwork setting. For instance, allowing agents the autonomy to decide whether to engage in the discussion or simply contribute to the final output.

4. Measuring final team performance by randomly querying a single agent within the team is a problematic evaluation choice. This approach essentially evaluates a single LLM operating with a noisy discussion context generated by weaker versions of itself or by agents lacking privileged information. Since most contemporary reasoning models are optimized for Chain-of-Thought (CoT) reflection processes, redundant multi-agent discussion effectively becomes noise, diluting useful signals (e.g., privileged information becomes too distant within the long context window). A more appropriate, team-oriented metric would compare the number of correct agent answers before (individual) and after (team) the discussion phase.

5. A general consensus within the multi-agent research community is that LLM systems benefit significantly from diversity in capabilities and perspectives. Previous research [1] has demonstrated that prompting agents to maintain distinct personas or utilize specific capabilities (e.g., tool calling, code generation) significantly improves overall performance compared to single-agent baselines. This raises concerns regarding the evaluation's validity across the broader spectrum of LLM agent frameworks, which often employ varied prompt engineering techniques (e.g. CoT, ReAct), complex interaction pipelines (e.g., verification, debate, voting mechanisms), and advanced base model capabilities. The current findings may not generalize well to heterogeneous teams equipped with diverse skill sets. Consider include those common multi-agent system designs as the benchmark baselines.

[1] Chen, Yongchao, et al. "Tumix: Multi-agent test-time scaling with tool-use mixture." arXiv preprint arXiv:2510.01279 (2025).

---

> ### Author Rebuttal · Authors · 2026-03-31
>
> We thank the reviewer for engaging with our work and for recognizing the "pressing yet underexplored research question" and "solid qualitative explanation." We address each concern below, after summarizing new baselines we ran in response to feedback.
>
> **Terminology.** ALOC (At Least One Correct) is the per-problem upper bound — the team gets credit if any member answered correctly; RE (Reveal Expert) = team told who the expert is; ENM (Expert Not Mentioned) = no expertise info shared (Section 3.3).
>
> **New baselines (conducted on HLE Text-Only due to time constraints):** Debate = Du et al. (2024) parallel broadcast-and-revise protocol, 4 agents, 4 rounds; Opt-Out = agents may abstain from discussion rounds; CoT+MV = independent chain-of-thought reasoning, majority vote, no interaction.
>
> | Method | Accuracy |
> |--------|:--------:|
> | ALOC (upper bound) | 45% |
> | Team (RE) | 34% |
> | Opt-Out (RE) | 31% |
> | Opt-Out (ENM) | 31% |
> | Team (ENM) | 28% |
> | Debate (ENM) | 26% |
> | Debate (RE) | 23% |
> | CoT + Majority Vote | 14% |
>
> All methods fall 11-31pp short of ALOC.
>
> **W1: "Given the enormous amount of existing work demonstrating the superiority of multi-agent LLM systems and the success of Mixture-of-Experts (MoE) architectures… I am inclined to attribute these divergent observations to the specific evaluation methods."**
>
> MoE is a *single-model architecture* with learned routing over sub-networks — not a multi-agent system. Section 2.1 explains how our setting differs from prior multi-agent work (which typically uses model copies with role personas or fixed workflows). We also reference existing works studying multi-agent failures in our related work. Other reviewers highlight "well-designed experiments and interesting findings" and recognize we ask "an important question for multi-agent LLM systems." Our new experiments find the same effects across debate and opt-out protocols, addressing generalization concerns.
>
> **W2: "Is the information asymmetry manipulation genuinely effective? Ground truth may be memorized."**
>
> Figure 2 shows this directly: No Information (approx. 50 ranking error) is substantially worse than ENM (~29). This 21-point gap demonstrates the manipulation is effective — if ground truth were memorized, privileged information would not help. Appendix C (Figures 5–7) confirms this across all tasks.
>
> **W3: "LLM agents are constrained by a fixed speaking sequence… forces contributions even when agents lack expertise."**
>
> We appreciate your great suggestions. The reviewer is correct that agents speak sequentially (randomized per round; Section 3, Appendix A.1) with fixed discussion length. We added two baselines that directly address this (see summary table): **opt-out** (agents may abstain; 31% on HLE) and **debate** (simultaneous broadcast; 23–26%). Both are fundamentally different structures, yet fall far short of ALOC.
>
> **W4: "Measuring final team performance by randomly querying a single agent is problematic… evaluates a single LLM operating with noisy discussion context."**
>
> Thank you for raising this, we agree. We now report team accuracy as the **majority vote of all agents' post-discussion individual answers**. Updated results (old→MV, ENM/RE): MMLU Pro 85→86/89→89, GPQA 74→76/82→83, SimpleQA 50→51/54→54, HLE 29→28/35→34, MATH-500 67→63/73→71. Max shift: 4pp. We adopt majority vote going forward.
>
> **W5: "Consider including common multi-agent system designs as benchmark baselines."**
>
> Great suggestion, thank you. Please see the summary table: we added Du et al. (2024) debate (23–26%) and CoT + majority vote (14%). Both fall far short of ALOC (45%), confirming the failure is not protocol-specific. Our work complements fixed-workflow systems (MetaGPT, AutoGen) — we study coordination that must emerge from interaction (Section 2.1).
>
> **Key Question: "Why were SOTA models not tested on the psychological tasks?"**
>
> Psychology tasks allow controlled expertise manipulation using cost-effective models (10 seeds × 3 compositions × 4 conditions per task). ML benchmarks confirm the same trends with frontier models (GPT-5, o4-mini, Claude Sonnet 4.5, Claude Opus 4). This two-tier design is explained in Section 3.2.
>
> **Key Question: "How exactly are the communication logs and discussion histories stored and represented within the context window?"**
>
> We will expand Appendix A.1. Each agent maintains a growing context containing (1) their Phase 1 response, (2) all agents' shared initial opinions + collaborative goal (and in RE, the expert revelation), and (3) prior rounds appended sequentially. Speaking order is randomized per round. Context grows linearly with rounds × agents.
>
> **On Soundness.** The reviewer's rating of 1/4 appears anchored on W1 (MoE — a single-model architecture, not multi-agent, and generalization concerns) and W2 (memorization). We have conducted experiments to address these concerns. We ask the reviewer to reconsider, and thank them for helping strengthen the paper!

---

> > ### Author Rebuttal · Reviewer_6PDV · 2026-04-01
> >
> > Thank you for the rebuttal. Most of my technical concerns have been addressed, I will raise my score. However, I would like further clarification regarding W1, specifically the conditions under which the synergy gap occurs.
> >
> > Based on your response, it appears that multi-agent interaction is beneficial when utilizing fixed workflows or role-specific personas (as seen in prior work), but can be harmful in self-organizing teams with information asymmetry. Does this imply that structural design based on human prior knowledge is a prerequisite for successful multi-agent systems? Alternatively, do you believe this is a limitation that will be overcome by stronger models or new multi-agent post-training paradigms?
> >
> > Additionally, what are the real-world implications of manipulating information asymmetry as described in your study? Specifically, do you expect your conclusions to generalize to decentralized environments or adversarial settings?

---

> > > ### Author Response · Authors · 2026-04-01
> > >
> > > Thank you for engaging further and for raising your score — we appreciate the thoughtful follow-up.
> > >
> > > **On structural design as a prerequisite**: our results suggest that self-organizing teams fail specifically at leveraging asymmetrically distributed expertise — not at collaboration in general. Fixed-workflow systems (AutoGen, Virtual Lab, MetaGPT, SiriuS) succeed partly because human designers pre-specify who contributes what, effectively solving the expertise identification and deference problem by construction. Our findings suggest that without such structure, current LLMs default to consensus-seeking behavior (integrative compromise, Section 4.5) rather than epistemic deference. Whether this is a prerequisite or a current limitation is an open question we find very interesting. We believe stronger models or multi-agent post-training could help — the failure may partly stem from RLHF incentivizing agreeable, consensus-building responses (though this remains a hypothesis, as noted in our revision). The fact that the gap persists across all frontier models we tested (including GPT-5 and Claude Opus 4) suggests it is not trivially resolved by scale alone, but targeted training for epistemic deference in group settings is a promising direction we hope our work motivates.
> > >
> > > **On real-world implications**: information asymmetry is ubiquitous in practice — medical teams (specialist vs. generalist), software teams (domain expert vs. generalist engineer), legal teams, etc. Our controlled manipulation mirrors any setting where team members have unequal domain knowledge. We would expect the expertise leveraging failure to be worse in decentralized settings, where there is even less information about who holds expertise. Interestingly, the same consensus-seeking behavior that hinders expertise leveraging may provide robustness in adversarial settings by outvoting adversarial input (Section 4.4, Appendix F) — exploring this tradeoff is promising future work.
> > >
> > > We would be happy to expand our discussion section to cover these topics (structural design as prerequisite vs. current limitation, real-world generalization, and the robustness tradeoff) if the reviewer would find that an improvement to the paper.

---

### Official Review · Reviewer_NA2P · 2026-03-12

**Soundness:** 3
**Presentation:** 3
**Significance:** 3
**Originality:** 3
**Overall Recommendation:** 5
**Confidence:** 3

**Summary:**

This paper investigates whether unconstrained, self-organizing multi-agent LLM teams can achieve "strong synergy" of matching or exceeding their best individual member's performance. The authors outline a notable area that most multi-agent LLM work hard-codes roles, workflows, or aggregation rules, so it is unclear whether genuinely free-form deliberative teams can achieve “strong synergy,” i.e., match or exceed their best member. The paper evaluates this question on classic organizational-psychology tasks (NASA Moon Survival, Lost at Sea, Student Body President) and several modern ML benchmarks (MMLU Pro, GPQA Diamond, HLE, MATH-500, SimpleQA), using 4-agent teams with multi-round discussion under settings where expertise is hidden, revealed, concentrated, or distributed. The central claim is that such teams consistently fail to match the best individual member, even when told who the expert is, and that the main bottleneck is not expert identification but expert leveraging.

**Compliance With Llm Reviewing Policy:**

Affirmed.

**Final Justification:**

While I appreciate the authors' substantial efforts during rebuttal, my core methodological concern remains unresolved: the inconsistent operationalization of "expertise" across task types undermines the paper's central claims about expert leveraging. In the psychology tasks, expertise is cleanly induced through privileged information, a controlled manipulation with clear causal interpretation. In contrast, for ML benchmarks, "expert" is retrospectively defined as whichever model happened to answer correctly, conflating expertise with outcome and preventing clean causal inference about leveraging failures. This asymmetry means the paper's cross-task generalization rests on conceptually incompatible constructs, weakening the theoretical coherence that the organizational psychology framing aims to provide. I would like to keep my score.

**Key Questions For Authors:**

1. Can you justify generalizing from your specific protocol to "LLM teams" broadly?
2. Why were these particular model compositions chosen for each benchmark? Was this prospective?
3. How do you reconcile the different operationalizations of "expert" across task types?

**Limitations:**

I would strongly suggest to add an impact statement

**Strengths And Weaknesses:**

Soundness of Technical Claims and Methodology
plus:
-I value the well-designed experiments spanning controlled psychology tasks and realistic ML benchmarks
-The negative results appear consistent across many conditions that the authors include, strengthening reliability
-Moreover, the authors attempt to provide mechanistic insights into why teams fail, going beyond surface-level observations

minus:
-However, the entire study depends on one specific discussion protocol (4 agents, 4 rounds, random final answerer, open group chat, prompt-engineered reveal instructions). This is a single design point, yet the paper repeatedly generalizes findings to "LLM teams" in general. I would argue that this overgeneralization is not justified
-Results are based on only 100 problems and 2 seeds, which is comparatively weak evidence for the strong claims made and which is insufficient statistical rigor for ML benchmarks.
-Table 2 lacks confidence intervals
-No hypothesis tests reported
-What I find problematic ist that the authors do not provide a consistent "expert" definition. In human tasks, expertise is induced via privileged information (clear operationalization), however in ML benchmarks, "expert" becomes the model that got the problem right, or the highest-accuracy model. I would argue that this is retrospective labeling and conceptually slippery and this inconsistency undermines cross-task comparisons
-Team compositions vary across benchmarks "to ensure variance in individual model performance". Why these specific four models for each benchmark? Was the selection prospective or post-hoc? Were alternatives tried and omitted? This raises concerns about cherry-picking

Presentation and Contextualization
plus:
-I value the framing through organizational psychology and the "strong synergy" criterion (relative to the best team member), which is useful and provides helpful conceptual grounding
-The authors provide a clear articulation of the core negative finding

minus:
-However, the paper speculates about RLHF/alignment as the cause of failure modes but provides no direct evidence
-Important methodological details are underspecified, reducing reproducibility. Claims about "self-organizing multi-agent teams" extend well beyond what the specific protocol can support

Significance of Contribution
Plus:
-This paper addresses a fundamental and timely topic with significant practical implications. It is valuable to document that naive free-form multi-agent discussion can underperform the strongest member, especially given the current hype around multi-agent systems
-The observation that larger teams can worsen performance is practically relevant and challenges the simplistic "more agents = better" narrative

Minus:
-	However, the paper mostly documents a failure mode under one specific protocol without proposing solutions or alternative designs, which provides limited actionable guidance for practitioners seeking to improve multi-agent collaboration

Originality
Plus:
-I value the novel framing through an organizational psychology lens, and the specific "strong synergy" criterion provides a useful evaluation standard
-I value the systematic benchmark organization across task types


Overall suggestions for revision
1.I would suggest adding statistical rigor by using confidence intervals, hypothesis tests, and more seeds
2.I would suggest clarifying and unifying the "expert" definition across benchmarks
3.I would suggest providing transparency on the model selection decision

Typos:
-PSCYHOLOGY (line 187, left column) is misspelled in a subsection heading

---

> ### Author Rebuttal · Authors · 2026-03-31
>
> We thank the reviewer for their thorough and constructive review, and for recognizing the value of our experimental design, mechanistic insights, and practical relevance. We address the reviewer's concerns after introducing terminology and new experiments we ran in response to reviewer feedback.
>
> **Terminology.** ALOC (At Least One Correct) is the per-problem upper bound — the team gets credit if any member answered correctly; RE (Reveal Expert) = team told who the expert is; ENM (Expert Not Mentioned) = no expertise info shared (Section 3.3).
>
> **New baselines (conducted during rebuttal on HLE Text-Only):** Debate = Du et al. (2024) parallel broadcast-and-revise protocol, 4 agents, 4 rounds; Opt-Out = agents may abstain from discussion rounds; CoT+MV = independent chain-of-thought reasoning, majority vote, no interaction.
>
> | Method | Accuracy |
> |--------|:--------:|
> | ALOC (upper bound) | 45% |
> | Team (RE) | 34% |
> | Opt-Out (RE) | 31% |
> | Opt-Out (ENM) | 31% |
> | Team (ENM) | 28% |
> | Debate (ENM) | 26% |
> | Debate (RE) | 23% |
> | CoT + Majority Vote | 14% |
>
> All methods fall 11-31pp short of ALOC.
>
> **"The entire study depends on one specific discussion protocol… this overgeneralization is not justified."**
>
> Thank you for pointing this out — we have addressed it with new experiments (see summary table): debate, opt-out, and CoT+MV cover four fundamentally different coordination schemes, all falling well short of ALOC. We will soften language in the revision, but the consistency across protocols substantially strengthens generalizability.
>
> **"Results are based on only 100 problems and 2 seeds… Table 2 lacks confidence intervals."**
>
> Thank you for raising this. Table 1 (psychology tasks) already reports mean ± SEM over 30 seeds, providing strong statistical grounding. For ML benchmarks, the 100-problem subsample across 2 seeds reflects the computational cost of full multi-agent team discussions with frontier models (4 agents × 4 rounds × 2 conditions per problem). We will run additional seeds and add confidence intervals to Table 2 in the revision. That said, the effects we report are large (8–38% relative synergy gaps) and consistent across all 8 tasks, both expertise distributions, and all team compositions — this consistency across conditions provides converging evidence beyond what any single benchmark's sample size would yield.
>
> **"The authors do not provide a consistent 'expert' definition… retrospective labeling."**
>
> Thank you for raising this. For psychology tasks, expertise is induced via privileged information — a clean, controlled operationalization. For ML benchmarks, the expert is the per-problem best-performing model. Crucially, in the RE condition we reveal expert identity to the team — the team has the information needed to defer, making the comparison to the expert baseline valid. This mirrors real-world settings where one may know who has domain expertise. We will clarify this in the revision.
>
> **"Team compositions vary across benchmarks… concerns about cherry-picking."**
>
> Team compositions were selected prospectively to ensure variance in individual model performance on each benchmark, allowing us to test expertise leveraging with genuine performance differentials. The specific compositions are documented in Appendix D. We will add language clarifying the prospective selection criterion.
>
> **"The paper speculates about RLHF/alignment as the cause… no direct evidence."**
>
> We agree this is speculative and will reduce emphasis in the main text, framing it as a hypothesis for future investigation rather than a demonstrated cause.
>
> **"The paper mostly documents a failure mode… limited actionable guidance for practitioners."**
>
> We appreciate this observation. Our new results demonstrate that the failure mode persists across alternative protocol designs (debate, opt-out), which itself is actionable: it tells practitioners that simply switching interaction protocols will not resolve the expertise leveraging problem. Additionally, our mechanistic analysis — decomposing the failure into identification vs. leveraging (Table 1) and identifying integrative compromise as the correlated behavioral pattern (Section 4.5) — provides concrete design targets: future systems should prioritize expertise when expertise asymmetries exist. We agree that developing solutions is important future work and will frame this more explicitly in the revision.
>
> **Impact statement.** We will add an impact statement in the revision: Our findings caution practitioners against assuming that multi-agent LLM teams will outperform individual experts, particularly in domains where expertise is asymmetrically distributed (e.g., medical diagnosis, legal analysis, financial decision-making). Deploying multi-agent teams without expertise-aware coordination mechanisms may systematically underperform a single well-chosen expert model, with implications for high-stakes applications.
>
> We thank the reviewer for helping strengthen the paper!

---

> > ### Author Rebuttal · Reviewer_NA2P · 2026-04-02
> >
> > Thank you for the rebuttal. Most of my concerns have been addressed.
> >
> > However, the authors have not fully reconciled the inconsistent definition of "expert" across tasks or provided complete statistical rigor for the benchmark results, leaving a methodological question open:
> > To conclusively isolate the effects, would a more controlled experimental design be required? Specifically, would it be more suitable to construct ML benchmark tasks where expertise is proactively engineered via asymmetric, privileged information (as in your psychology tasks), rather than inferred from model performance?

---

> > > ### Author Response · Authors · 2026-04-02
> > >
> > > Thank you for the thoughtful follow-up.
> > >
> > > **On the expert definition:** This is an excellent point. Our two-tier design was intended to provide exactly this complementarity: psychology tasks offer controlled, proactively engineered expertise manipulation (privileged information creates a known expert), while ML benchmarks test whether the same patterns hold "in the wild," where expertise asymmetry arises naturally from inherent model differences. We view these as complementary — the psychology tasks let us study the expertise leveraging phenomenon from organizational psychology in a controlled setting, and the ML benchmarks demonstrate that the same phenomenon appears under naturalistic conditions.
> > >
> > > The reviewer raises an interesting middle ground: constructing ML-style tasks where expertise is induced (e.g., providing a relevant code snippet or retrieved passage to one agent). This could bridge the controlled and naturalistic settings, and we agree it is a promising direction for future work. We would be happy to discuss this in the revision.
> > >
> > > **On statistical rigor:** We appreciate the push here. Table 1 (psychology tasks) already reports mean ± SEM over 30 seeds, providing strong statistical grounding. For Table 2 (ML benchmarks), each seed requires up to 8 hours of frontier model inference per benchmark per condition. We hope that the consistency of the effect across all 8 tasks, 2 expertise distributions, multiple team compositions, and now 4 coordination protocols (our deliberation protocol, Du et al. 2024 debate, opt-out, and CoT + majority vote) provides strong converging evidence. We will run additional seeds and report standard deviations for the final revision, but this is difficult to accomplish within the rebuttal window.
> > > We appreciate the reviewer's continued engagement in strengthening the paper.

---

### Official Review · Reviewer_oJ5g · 2026-03-13

**Soundness:** 2
**Presentation:** 4
**Significance:** 3
**Originality:** 2
**Overall Recommendation:** 4
**Confidence:** 4

**Summary:**

This paper studies the effects of multi-team agents with a psychological framing. It claims that multi-agent teams are not ideal and do not achieve the best possible performance. It analyzes this across two settings---NASA survival scenarios and ML Benchmarks. It then claims that this is due to the fact that agents strive for compromise rather than deferring to the expert.

**Compliance With Llm Reviewing Policy:**

Affirmed.

**Final Justification:**

Satisfied some of my most major concerns around rigor. Could still be improved, but now more of a borderline paper to me.

**Key Questions For Authors:**

Questions asked above.

**Limitations:**

yes

**Strengths And Weaknesses:**

Strengths:
- The paper is well written and easy to follow.
- The topic and framing is interesting.
- The experiments and results are also clear and interesting.
- I find the qualitative analysis particularly useful and well explained.

Again, I think the overall paper is well written and I can see the core of a strong paper, but the following weaknesses hold it back.

Weaknesses:

I will split my weaknesses into two portions as I believe there is a core fundamental weakness to this paper that is difficult to address without significant reframing.

Main Weakness
-----
This work claims in the title and throughout the text that multi-agent teams hold the expert back. This is a strong claim, and one that I believe does not match the experimental results, especially once you dig beyond the main paper and into the results provided in the appendix.

For the NASA survival experiments, it's unclear to me if this is just due to noise as even in the expert only scenario where they are given ground-truth information it does not achieve perfect performance! In both model families, there's always error in ranking. Thus, if you simply add more models (even if they were all experts), I would expect that performance would degrade. This experiment (4 expert models) is one I would like to see as a baseline to convince me otherwise.

In the ML experiments, it confuses me why the main paper focuses on at-least-one-correct solely rather than the "best individual model" which is studied in the appendix. at-least-one-correct is a strangely high bar to compare against and not achieving this is not inherently anti-synergistic to me. It seems that in all the benchmarks in fact improve when put together as a team and beat the "best individual model" consistently (which the authors even agree with). To confirm my understanding, I read Bonner et. al, 2002 and find that even in their experiments they have one individual that is the best performer. This would better match the "best individual model" experiment rather than at-least-one-correct.

This is all not to say that multi-team collaboration is perfect. In fact, I agree with the experimental results that there is a gap. But to claim that teams "hold the expert back" seems a misleading claim from the experimental results and fundamentally weakens the paper. This is also not a minor reframing, hence (combined with my other weaknesses) my score.

Other Weaknesses (labeled minor or not)
-----
- How are the prompts created? They are claimed to be "aggressively optimized" prompts, yet there is no experiment here. Since the core of this paper is so prompt dependent, I believe this is something that must be clearly addressed. For example, you could have easily had a prompt that simply states that X agent is the expert and **is correct in its answer**. This would likely lead to improved synergy in the expert revealed scenario! One could claim that this is perhaps an overly aggressive prompt, but my point is that it's not clear where that line lies and why the prompt in the paper was chosen for any reason other than an arbitrary decision by the authors.
- Also, it's a bit unclear to me the novelty of this work. Is it simply the framing? We already know that agents have a collaboration gap. What is the new information that this paper is providing?
- As mentioned in my main weaknesses, there are additional experiments that I would have liked to see to better understand agent behavior.
- [minor] It's unclear to me why you don't run all models for all experiments. The claim is to observe diversity, but would you not have more diversity if you ran every model across all experiments?
- [minor] Figure 5 and 6 in the appendix. Captions do not match the figure.

---

> ### Author Rebuttal · Authors · 2026-03-31
>
> We appreciate the reviewer's recognition that the paper is "well written and easy to follow" with "interesting" framing and "particularly useful" qualitative analysis, and we take their concerns seriously. We address each below, after summarizing experimental conditions and new baselines we ran in response to feedback.
>
> **Terminology.** ALOC (At Least One Correct) is the per-problem upper bound — the team gets credit if any member answered correctly; RE (Reveal Expert) = team told who the expert is; ENM (Expert Not Mentioned) = no expertise info shared (Section 3.3).
>
> **New baselines (conducted on HLE Text-Only due to time constraints):** Debate = Du et al. (2024) parallel broadcast-and-revise protocol, 4 agents, 4 rounds; Opt-Out = agents may abstain from discussion rounds; CoT+MV = independent chain-of-thought reasoning, majority vote, no interaction.
>
> | Method | Accuracy |
> |--------|:--------:|
> | ALOC (upper bound) | 45% |
> | Team (RE) | 34% |
> | Debate (RE) | 23% |
> | Opt-Out (RE) | 31% |
> | Team (ENM) | 28% |
> | Debate (ENM) | 26% |
> | Opt-Out (ENM) | 31% |
> | CoT + Majority Vote | 14% |
>
> All methods fall 11-31pp short of ALOC.
>
> **Main Weakness: "At-least-one-correct is a strangely high bar... teams beat the best individual model consistently."**
>
> ALOC is directly grounded in the organizational psychology literature. Bonner et al. (2002) — which the reviewer consulted — defines strong synergy as teams matching their best member on each problem. ALOC operationalizes exactly this: which model is the best member changes per problem, so this is a per-problem measure. Human teams *do* achieve this when expertise is revealed.
>
> Regarding the title: we are open to adjusting it — for example, "Multi-Agent Teams Hold Experts Back: LLM Collaboration Fails to Leverage Expertise" or a similar formulation that retains the core message while clarifying scope. We welcome the reviewer's guidance on a title they feel is appropriate.
>
> **"If you simply add more models (even if they were all experts), I would expect performance would degrade."**
>
> Thank you — this prompted us to include a previously omitted baseline. We ran a team full information condition in which all 4 agents receive complete expert knowledge. Results on the psychology tasks:
>
> | Task | Single Expert | Full Info Team (4 experts) |
> |------|:------------:|:-------------------------:|
> | NASA Moon Survival | 15.03 ± 0.87 | 16.22 ± 1.99 |
> | Lost at Sea | 21.38 ± 0.87 | 19.11 ± 2.23 |
> | Student Body President | 2.60 ± 0.31 | 1.78 ± 0.71 |
>
> When all agents have expertise, team performance matches the single expert. The collapse occurs specifically under expertise asymmetry — not communication overhead, but failure to leverage asymmetrically distributed expertise.
>
> **"How are the prompts created? ... you could have had a prompt that simply states that X agent is the expert and is correct in its answer."**
>
> Our RE prompt (Appendix A.3) already aggressively instructs deference: "Adopt Agent {X}'s COMPLETE ranking as your baseline... Do NOT prioritize your own reasoning." The gap persists despite this. We would appreciate feedback if the reviewer believes the prompt could be stronger.
>
> Telling the team someone is *correct* would invalidate the design: at test time, one may know who has expertise but never who is actually correct. We therefore do not reveal accuracy information.
>
> **"It's unclear to me the novelty of this work. We already know that agents have a collaboration gap."**
>
> Section 2.1 addresses this directly. Much of the prior multi-agent literature — Du et al. (2024), Hong et al. (2024), and others — reports performance *improvements* from collaboration. Our five novelty axes are: (1) genuine model heterogeneity, (2) deliberative interaction, (3) no pre-specified roles/workflows, (4) strong synergy criterion, (5) org-psych grounding. Other reviewers highlight "well-designed experiments and interesting findings" and praise the "careful behavioral analysis." The identification vs. leveraging decomposition (Table 1) provides actionable insight absent from prior work.
>
> **"[minor] Why don't you run all models for all experiments?"**
>
> Psychology tasks require controlled expertise manipulation across 10 seeds x 3 team compositions x 4 information conditions per task (360+ runs). Frontier models would be prohibitively expensive for this volume. ML benchmarks use frontier models to test practical potential. The two-tier design is explained in Section 3.2.
>
> **"[minor] Figure 5 and 6 captions do not match the figure."**
>
> We will update the figure captions to note variance in the leveraging gap by team composition while clarifying the overall trend remains.
>
> We believe we have addressed all concerns with new results and ask that the reviewer reconsider their assessment. We again thank the reviewer for engaging with the work and helping us strengthen the paper!

---

> > ### Author Rebuttal · Reviewer_oJ5g · 2026-04-04
> >
> > Thank you for responding. Unfortunately, I am still not convinced that this paper is ready to be accepted. My main concerns are still that the claims are incredibly strong, but the results are not strong enough to match those claims.
> >
> > First (Bonner / ALOC)
> > -----
> > In reference to Bonner et al. (2002), the experiment tests participants on Mastermind problems prior to the task to choose the expert. What is important is that this defines expertise based on **problem domain** not a **specific** mastermind problem. The expert in each Mastermind 3-group is not doing the same exact problem they did before (note that they even do 2 trials to determine the expert in the paper).
> >
> > The equivalent experiment in your situation would be to choose a problem domain within each benchmark and split each domain into problems that are used to determine the expert vs problems that are held-out to conduct your experiments. You are setting a much higher bar on what successful collaboration means by assigning expertise based on problems that you know the model is able to solve correctly.
> >
> > Second (all experts)
> > ----
> > Thank you. This answers my question. For reference, I was referring to how collaboration has been shown to be difficult already (See: https://arxiv.org/abs/2511.02687). This helps convince me that expertise has something to do with your results rather than simply collaboration.
> >
> > Third (prompts)
> > -------
> > The choice of prompt is clearly **critical** to your experimental results. Yet there are no experiments detailing how your prompt came to be. There is a wide range of potential prompts between what you wrote and what I wrote (which you describe as not feasible for experimentation). That your prompt "aggressively instructs deference" and my prompt "would invalidate the design" is simply a claim that you are making without rigorous experimental results to back that up.
> >
> > In my opinion, this: "Do NOT prioritize your own reasoning over the expert’s. The goal is to preserve the expert foundation while incorporating any genuinely valuable insights. Avoid compromising or averaging—either keep the baseline item
> > position, or make a clearly justified change." is simply NOT aggressive enough. You think my previous suggestion would invalidate the experiment, but I could easily see a statement such as "You should assume that the expert is correct in their judgement." or some other statement changing your results. I am not saying this is correct, but simply *claiming* that your prompt is "aggressive" enough is not rigorous. Once again, bold claims require equally rigorous results.
> >
> > For example you could run GEPA on your prompt optimizing for performance. I am confident at some point it will optimize your prompt in such a way that your current results are invalidate as it will improve performance overall. You need to show that such an optimization does not invalidate your results (or explain rigorously why such optimizations lead to prompts that are incorrect/reward-hacked).
> >
> > Four (novelty)
> > ----
> > Makes sense, but leaving this paper again (https://arxiv.org/abs/2511.02687)
> >
> > The last two responses are minor and the responses are acceptable.
> >
> > The title is not the issue. It is bold and I actually quite like that. However, with such a bold title I expect equally rigorous experimentation. I do not feel that the paper currently reaches that level of rigor. Unfortunately, I cannot raise my score due to what I believe is a lack of rigor in the experimentation. Additionally, to be clear, this is only one aspect of my reasoning. Even if the claims were weaker, I still feel like the lack of rigor in some experiments would lead me to a similar score.
> >
> > EDIT
> > -----
> > These GEPA experiments are great and exactly what I'm looking for. I would be interested to read the prompts that are generated. Appendix is fine.
> >
> > Just to be clear, I am not associated with that paper (that's just the one I know of that has done this work); I just genuinely think this work is interesting, and want to see it improved.
> >
> > That said, with the GEPA experiments and all expert experiments have significantly increased the rigor in my opinion. There's still gaps, but I think I can bump it up to a weak accept.

---

> > > ### Author Response · Authors · 2026-04-07
> > >
> > > Thank you for the continued engagement. We ran the prompt optimization experiment you suggested.
> > >
> > > **GEPA prompt optimization (5 rounds per task).** We ran GEPA (Agrawal et al., 2025; ICLR 2026 Oral) on both NASA Moon Survival and HLE Text-Only. For HLE, GEPA was optimized on a 10-problem subset and evaluated on the full 100 problems due to cost considerations.
> > >
> > > | HLE Condition | Accuracy |
> > > |-----------|:--------:|
> > > | RE (original prompt) | 35% |
> > > | RE (GEPA-optimized) | 36% |
> > > | Expert alone (ALOC) | 51% |
> > >
> > > | NASA Condition | Mean L1 (↓) |
> > > |-----------|:-------:|
> > > | RE (original prompt) | 23.2 |
> > > | RE (GEPA-optimized) | 16.8 |
> > > | Expert alone | 13.6 |
> > >
> > > In both cases, GEPA converged to extreme-deference prompts. On HLE, this yields no meaningful improvement — the team still falls 15pp short of ALOC. On NASA, the gap narrows, but the optimized prompt converges to suppressing deliberation entirely.
> > >
> > > Due to character limits we will include the full GEPA-optimized prompts in the appendix in the final revision, but to summarize: the NASA prompt includes "PRIORITY RULE (NON-NEGOTIABLE): Treat Agent {X}'s ranking as the correct answer by default. The safest strategy is to copy Agent {X}'s ranking exactly," instructs "EXTREME CAUTION ABOUT CHANGES" requiring all of four conditions be met before any modification, states "Expect very few changes. Never rebuild from scratch," and concludes "Your final team ranking should usually match Agent {X}'s ranking exactly."
> > >
> > > Even with automated prompt optimization, the expertise leveraging gap persists: the team either fails to match the expert (HLE) or approaches it only by ceasing to function as a team (NASA). The reviewer predicted that GEPA would "optimize your prompt in such a way that your current results are invalidated." We ran this experiment and the results were not invalidated — they were reinforced. We believe this addresses the prompt sensitivity concern.
> > >
> > > We thank the reviewer for the suggestions — they have strengthened the paper. We will also cite Davidson et al. (2025) in the revision.

---

### Decision · Program_Chairs · 2026-04-30

**Decision:**

Accept (regular)

**Comment:**

The paper studies whether multi-agent LLM teams can match their best individual member, and shows a consistent gap, attributed to poor expertise leveraging. Reviewers generally find the paper clear and the results interesting.

Summary of concerns.
- The main concern is about the over-strong claims and limited generalization beyond the specific protocol.
- There are also some issues around the definition of "expert" and prompt sensitivity.
- Statistical rigor on benchmarks is also somewhat limited.

The rebuttal adds useful experiments (new baselines, majority vote, prompt optimization), which address several concerns and improved the paper quality, but some conceptual and rigor gaps remain.

Overall, this is a solid but somewhat over-claimed and borderline paper.